# Somatostatin-positive interneurons in the dentate gyrus of mice provide local- and long-range septal synaptic inhibition

**Mei Yuan[1,2†], Thomas Meyer[1†], Christoph Benkowitz[1], Shakuntala Savanthrapadian[1], Laura Ansel-Bollepalli[3], Angelica Foggetti[3], Peer Wulff[3], Pepe Alcami[1], Claudio Elgueta[1], Marlene Bartos[1*]**

[1]Systemic and Cellular Neurophysiology, Institute for Physiology I, University of Freiburg, Freiburg, Germany; [2]Faculty for Biology, University of Freiburg, Freiburg, Germany; [3]Institute for Physiology, University of Kiel, Kiel, Germany

**Abstract** Somatostatin-expressing-interneurons (SOMIs) in the dentate gyrus (DG) control formation of granule cell (GC) assemblies during memory acquisition. Hilar-perforant-path-associated interneurons (HIPP cells) have been considered to be synonymous for DG-SOMIs. Deviating from this assumption, we show two functionally contrasting DG-SOMI-types. The classical feedback-inhibitory HIPPs distribute axon fibers in the molecular layer. They are engaged by converging GC-inputs and provide dendritic inhibition to the DG circuitry. In contrast, SOMIs with axon in the hilus, termed hilar interneurons (HILs), provide perisomatic inhibition onto GABAergic cells in the DG and project to the medial septum. Repetitive activation of glutamatergic inputs onto HIPP cells induces long-lasting-depression (LTD) of synaptic transmission but long-term-potentiation (LTP) of synaptic signals in HIL cells. Thus, LTD in HIPPs may assist flow of spatial information from the entorhinal cortex to the DG, whereas LTP in HILs may facilitate the temporal coordination of GCs with activity patterns governed by the medial septum.

*For correspondence: marlene.bartos@physiologie.uni-freiburg.de

[†]These authors contributed equally to this work

## Introduction

The DG is situated between the entorhinal cortex and the CA3 area of the hippocampus, forming the first stage of the classical trisynaptic circuit (*Andersen et al., 1971*; *Eichenbaum, 1993*; *Lisman, 1999*). Together with the hippocampus, it plays an indispensable role in the formation of new memories and memory associations in various species including humans, nonhuman primates and rodents (*Burgess et al., 2002*; *Leutgeb et al., 2005*; *Buzsáki and Draguhn, 2004*; *Bakker et al., 2008*). It receives a rich multimodal input from the entorhinal cortex via the perforant path which carries information on various modalities of external cues and translates the rich input stream into sparse segregated ('orthogonalized') representations, a process called pattern separation (*Marr, 1971*; *Treves and Rolls, 1994*; *Leutgeb et al., 2007*; *Kitamura et al., 2015*). By separating the rich input stream into non-overlapping memories, the DG allows a high resolution of information (*Marr, 1971*). Consistent with this theory, GC activity is sparse (*Liu et al., 2012*; *Ramirez et al., 2013*; *Denny et al., 2014*; *Danielson et al., 2016*) and governed by strong GABAergic inhibition (*Nitz and McNaughton, 2004*; *Pernía-Andrade and Jonas, 2014*).

Which interneuron types may contribute to the sparse activity in the DG circuitry? Synaptic inhibition is provided by two main interneuron types in the hippocampus, parvalbumin-expressing perisoma-inhibiting fast-spiking interneurons (PVIs) and somatostatin-expressing dendrite-inhibiting cells (SOMIs; *Freund and Buzsáki, 1996*; *Rudy et al., 2011*). Their functional role in the DG markedly depends on their morphological properties and the location of their output synapses. PVIs receive

convergent excitatory inputs from the perforant path at their apical dendrites and from local GCs via their mossy fiber synapses at their basal dendrites and provide powerful feedforward and feedback inhibition to large populations of GCs (*Sambandan et al., 2010*). Rapid and powerful synaptic inhibition plays an important role in the timing of GC discharges (*Jonas et al., 2004*; *Sambandan et al., 2010*). Precise spike timing has been proposed to be of particular importance for the encoding of information as well as the generation of fast synchronous network oscillations (*Cobb et al., 1995*; *Buzsáki and Draguhn, 2004*; *Bartos et al., 2007*). Much less information is available on the functional integration and computational role of DG-SOMIs. HIPP cells have so far been considered to be synonymous for DG-SOMIs (*Freund and Buzsáki, 1996*; *Mott et al., 1997*). Indeed, immunohistochemical investigations showed that SOM-positive axon fibers predominantly project in the outer half of the molecular layer, co-aligned with the perforant path (*Halasy and Somogyi, 1993*; *Han et al., 1993*; *Mott et al., 1997*; *Hosp et al., 2014*; *Savanthrapadian et al., 2014*). Moreover, electron microscopical studies revealed SOM-expressing terminals at apical dendrites of GCs and very likely interneurons in the DG (*Leranth et al., 1990*; *Peng et al., 2013*). Thus, HIPP cells are in the optimal position to control information flow from the entorhinal cortex via the DG to the CA3 area by potentially influencing dendritic computation and synaptic plasticity (*Miles et al., 1996*; *Maccaferri, 2005*). PVIs and SOMIs are therefore ideally suited to control information processing in neuronal networks in a complementary manner (*Méndez and Bacci, 2011*).

However, very little is known on how SOMIs are functionally integrated in the DG neuronal network. Moreover, how can the apparent paradox of HIPP morphology and dense hilar SOM-expressing axon projections be reconciled (*Leranth et al., 1990*; *Peng et al., 2013*)? Here, we addressed these fundamental questions by applying single and paired whole-cell patch-clamp recordings of SOMIs in acute slice preparations of the rodent DG in combination with intracellular labeling and their optogenetic recruitment. We first provide evidence that DG-SOMIs fall in at least two contrasting types with distinct morphological and functional properties. We show that HIPP cells (*Halasy and Somogyi, 1993*; *Han et al., 1993*; *Mott et al., 1997*; *Hosp et al., 2014*) are classical feedback inhibitory interneurons providing weak and slow dendritic-inhibition onto GCs and local interneurons. We identified a new DG-SOMI type, hilus-associated interneurons (HILs), with axon collaterals in the hilus. They exert strong perisomatic inhibition onto local GABAergic inhibitory cells. Moreover, HIL but not HIPP cells form long-range connections to the medial septum. The functional integration of DG-SOMIs into local and long-distance neuronal networks places these cells in an ideal position to regulate sparse coding of spatial information forwarded by the entorhinal cortex to the DG, and to synchronize this circuit computation with theta activity patterns driven by the medial septum. Such coordination would be of particular importance during navigation when spatial information processing must be temporally coordinated with running velocity.

## Results

### Layer-specific axon distributions define two contrasting DG-SOMI types

To examine the morphological and physiological characteristics of DG-SOMIs, we performed whole-cell patch-clamp recordings of single GFP-expressing cells in acute hippocampal slice preparations of SOM-GFP transgenic mice (*Oliva et al., 2000*; *Figure 1A*). To validate the specificity of GFP-expression, we used antibody labeling against SOM in perfused material (Materials and methods). SOM-expressing cells were visualized with secondary antibodies conjugated to Cy3 (SOM-Cy3). Confocal image stacks revealed that $94.1 \pm 2.5\%$ of GFP-labeled cells expressed SOM (SOM[+]; *Figure 1B*; three mice, six slices / mouse; dorsal and ventral DG). Consistent with previous reports in CA1 and CA3 (*Oliva et al., 2000*), only a fraction of SOM-Cy3-labeled cells co-localized GFP ($62.8 \pm 4.8\%$; *Figure 1B*). Thus, GFP-expression is a reliable marker for SOM[+] cells in the DG.

GFP[+] cells were filled with biocytin during recordings and visualized post-hoc with Alexa Fluor 647-conjugated streptavidin for subsequent morphological characterization (*Figure 1C*; *Figure 1—figure supplement 1*; Materials and methods). Light-microscopy of 39 examined cells revealed that cell bodies and the majority of the dendrites of DG-SOMIs were located in the hilus. However, the largest proportion of labelled cells had axon fibers either dispersed in the molecular layer (8 out of 39 cells) or in the hilus (24 out of 39 cells). The remaining SOMIs (7 out of 39) were morphologically variable and showed neither preference for axon distributions in the hilus nor in the outer molecular layer (

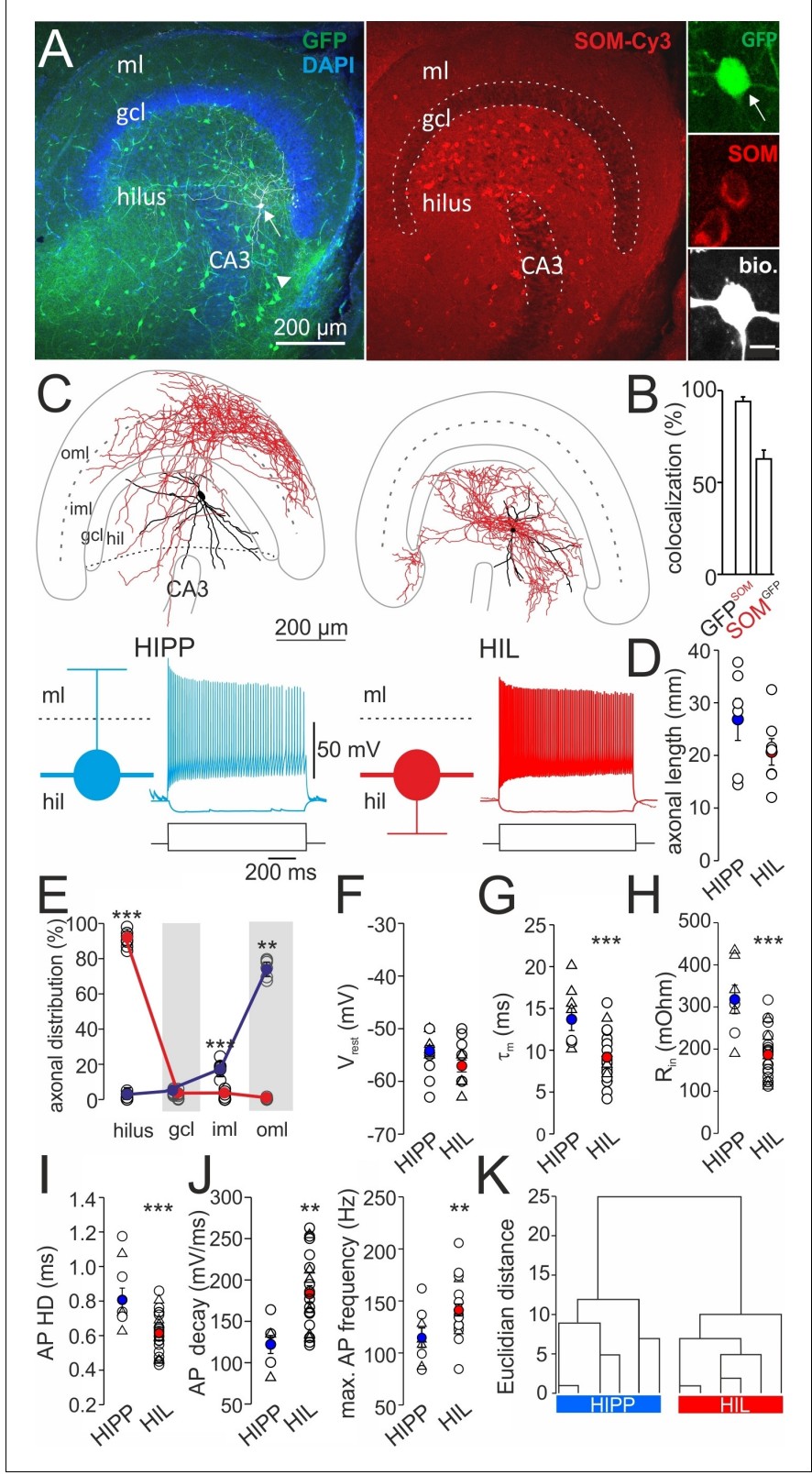

**Figure 1.** Two morphologically and physiologically contrasting DG-SOMI types. (**A**) *Left*, confocal image stack of a transverse section through the dentate gyrus (DG) of a transgenic mouse expressing GFP and somatostatin (SOM) in GABAergic interneurons (GIN; *Oliva et al., 2000*). Arrow, points to a GFP-positive cell intracellularly labelled with biocytin and conjugated to Alexa Fluor 647 (white cell). *Right*, same section showing antibody labelling

*Figure 1 continued on next page*

*Figure 1 continued*

against SOM. *Inset*, intracellularly labelled cell co-expresses GFP and SOM. Scale bar 5 μm. Arrow head points to areas of high SOM axon profile density. (**B**) Quantification of GFP and SOM co-localization (three mice). (**C**) Reconstructions of two representative intracellularly labeled DG-SOMI types. Somata and dendrites are depicted in black and axons in red. Grey lines depict layer-specific borders. *From left to right*, hilar perforant path-associated interneuron (HIPP), hilus-associated interneuron (HIL). *Below*, representative voltage traces of the two SOMI types during 1 s, −100 and 300 pA current injections. Schematics and color codes represent the respective SOMI types throughout all figures. Hilus is defined as area between the granule cell layer (gcl)-to-hilus border and the black striped line (***Freund and Buzsáki, 1996***). (**D**) Total axonal length of the two SOMI types on the basis of single-cell reconstructions in the DG (six cells each group). (**E**) Layer-specific axonal distribution of the two SOMIs in DG sub-areas. Red and blue circles connected by lines correspond to mean values of HIL and HIPP cells, respectively. (**F–H**) Summary plot of membrane resting potential ($V_{rest}$), membrane time constant ($\tau_m$) and input resistance ($R_{in}$) for both interneuron types. (**I,J**) Summary graph of the half-duration (HD) of single action potentials (APs), the decay of single APs and the maximal discharge frequency. (**K**) Hierarchical cluster analysis on the basis of morphological and physiological properties of 12 reconstructed cells (depicted by triangles in F-J) reveals two DG-SOMI classes which correspond to previously denominated HIPP and HIL cells (see Materials and methods). Circles represent single data points, circles with lines are means ± SEM; ***p≤0.001, **p≤0.01, *p≤0.05. Abbreviations: gcl, granule cell layer; hil, hilus; iml, inner molecular layer; oml, outer molecular layer.

The following figure supplements are available for figure 1:

**Figure supplement 1.** Morphological reconstructions of HIPP and HIL cells in the DG.

**Figure supplement 2.** Morphological reconstructions of non-HIPP and non-HIL cells of the DG.

**Figure supplement 3.** HIPP and HIL cells generate action potentials with different voltage trajectories.

*Figure 1—figure supplement 2*) and were therefore not further examined in this study. Detailed morphological reconstructions of a subset of labeled cells confirmed our initial observation. One group of DG-SOMIs had HIPP morphologies (***Halasy and Somogyi, 1993***; ***Han et al., 1993***; ***Hosp et al., 2014***; ***Savanthrapadian et al., 2014***) with axon fibers distributed in the molecular layer with highest proportion in the outer molecular layer (74.7 ± 2.1%; six reconstructed cells; ***Figure 1C,E***; ***Figure 1—figure supplement 1***). Some axon fibres projected in the hilus (1.7 ± 1.1%, ***Figure 1C,E***) consistent with our recent observation that HIPPs are interconnected by perisomatic synapses (***Savanthrapadian et al., 2014***). In marked contrast, the second group of recorded SOMIs projected almost exclusively in the hilus (91.0 ± 1.7% of the total axonal length; six reconstructed cells) and were therefore termed hilus-associated interneurons (HILs; ***Figure 1C***, right; ***Figure 1—figure supplement 1***). Both cell types largely avoided the granule cell layer (HIPP 3.9 ± 0.6%, HIL 2.1 ± 0.8% of the total axonal length; ***Figure 1E***). Despite the marked layer-specific axonal dispersion, the total dendritic and axonal length in the DG was similar among both types (dendrite: HIPP 2500.4 ± 333.0 μm, HIL 2547.3 ± 427.8 μm, p=0.935, *t*-test; axon: HIPP 26.8 ± 4.0 mm, HIL 20.7 ± 2.5 mm, p=0.205, *t*-test; ***Figure 1D***). Antibody labelling revealed that the majority of identified HIPP and HIL cells in this study fulfilled the criterion of SOM-expression (33 out of 35 cells; ***Figure 1A***, inset). Finally, both neuron types were morphologically and physiologically identified in SOM-tdT mice (SOM-Cre x Ai9-RCL-tdT; Materials and methods) confirming their strain-independent occurrence (seven light-microscopically identified HIPP and 7 HIL cells). Thus, the DG contains at least two morphologically contrasting SOMI types, HIPP cells with axon fibers dispersing predominantly in the outer molecular layer and HIL cells, with axon largely located in the hilus.

## DG-SOMI types have different intrinsic membrane properties

Next, we examined the passive and active membrane properties of the two DG-SOMI types (***Figure 1F–J***; see Materials and methods). Consistent with their comparable dendritic and axonal length in the DG, we observed no difference in the membrane capacitance (HIPP 47.8 ± 6.3 pF, 7 cells; HIL 50.5 ± 2.4 pF, 22 cells; p=0.63). The membrane resting potential was also similar (HIPP −54.2 ± 1.0 mV, 14 cells; HIL −57.1 ± 1.1 mV, 22 cells; p=0.079, *t*-test; ***Figure 1F***). However, HIPPs had a ~ 1.5 times slower membrane time constant ($\tau_m$) and a ~1.7 times higher input resistance than

HILs ($\tau_m$: HIPP 14.3 ± 1.4 ms, HIL 9.3 ± 0.6 ms; p<0.001; input resistance: HIPP 317.9 ± 34.0 MΩ, 8 cells; HIL 186.7 ± 12.1 MΩ, 24 cells; p<0.001; $t$-test; *Figure 1G,H*). Moreover, the half-duration of single action potentials was longer for HIPP than for HIL cells (0.86 ± 0.08 ms *vs* 0.62 ± 0.03 ms; p<0.001, $t$-test; *Figure 1I*; *Figure 1—figure supplement 3*). The narrower spike width in HILs was correlated with a larger slope in the decay of single action potentials (HIPP 122.0 ± 10.9 mV ms$^{-1}$, HIL 183.3 ± 9.8 mV ms$^{-1}$; p=0.003, $t$-test; *Figure 1J*, left) suggesting a different expression profile of voltage-gated channels in the two SOMI types. Furthermore, disparities were observed in activity patterns evoked by long-lasting positive current injections with step-wise increasing amplitudes. The maximal discharge frequency was lower for HIPP than for HIL cells (111.4 ± 10.2 Hz *vs* 141.5 ± 5.7 Hz; p=0.015, $t$-test; *Figure 1J*, right). Moreover, the ratio between the last and the first inter-spike-intervals at maximal discharge frequency was higher for HIPPs (2.1 ± 0.09) indicating a stronger adaptation of spike trains than for HIL cells (1.6 ± 0.12; p=0.004, Mann-Whitney $U$ test). Thus, DG-SOMIs show differences in their membrane characteristics favoring slow signaling in HIPP and rapid signaling in HIL cells.

To further test whether DG-SOMIs can be classified into independent types, we performed a hierarchical cluster analysis on the basis of morphological variables obtained from the fully reconstructed interneurons and their passive and active membrane characteristics (*Figure 1K*; depicted as triangles in *Figure 1F–J*; Materials and methods). We found that interneurons fell into two classes separated by an Euclidian linkage distance of 25% (*Figure 1K*). The first cluster was formed by slow signaling HIPP cells with axon collaterals largely located in the outer molecular layer, whereas the second cluster was formed by fast-spiking HIL cells with axon collaterals largely constrained to the hilus. Thus, the combination of morphological and physiological parameters allows the classification of DG-SOMIs into two distinct types.

## HIL but not HIPP cells form long-range connections to the medial septum

Previous tracing studies proposed that DG-SOMIs project to the medial septum (DG-septal cells; *Jinno and Kosaka, 2002*). To examine whether our set of identified SOMIs included long-range projecting DG-septal interneurons, we injected Cre-inducible rAAV vectors encoding GFP bilaterally in the dorsal DG of SOM-Cre mice (*Figure 2*; Material and methods). Cre-induced GFP-expression was highly specific as confirmed by antibody labeling against SOM (95.4 ± 3.2% co-localization; seven slices, three mice; *Figure 2A,C*). Moreover, GFP-expressing cell bodies were restricted to the hilus, defined as the area between the granule cell layer and the pyramidal cell layer of CA3 (see *Figure 1C* left, black dashed line), in line with earlier immunohistochemical reports (*Acsády et al., 2000*; *Peng et al., 2013*). GFP$^+$ axonal fibers were found in the hilus and the molecular layer but rarely in the granule cell layer confirming the spatial specificity of the DG injection site (*Figure 2A*).

Three-dimensional-images of clarity-processed whole mounts of injected brains (Materials and methods) showed that SOM$^+$ axon collaterals projected to the hippocampal fissure and along the fimbria to the medial septum and the vertical limb of the diagonal band of Broca (MSvDB; *Figure 2B*). Labeled axons in the MSvDB showed some variability in their appearance. They were either thick with few varicosities or thin with several boutons of different morphology (*Figure 2B*, inset). To identify the nature of DG-septal projecting SOMIs, we retrogradely labeled them by injecting a red fluorescent retrograde tracer (RedRetroBead) into the MSvDB (*Figure 2D*). After 3–8 days, we identified numerous red labeled cell bodies located in the hilus as well as in the stratum oriens and radiatum of CA1 and CA3 (26.5 ± 2.4% of SOM-expressing cells were labeled with RedRetroBead, 106 SOM cells; seven slices, two mice), confirming previous data on hippocampal-septal projecting SOMIs (*Jinno and Kosaka, 2002*; *Gulyás et al., 2003*). Colocalization analysis revealed that cell bodies of virtually all retrogradely tagged DG-septal projecting neurons expressed SOM (93.4 ± 2.2%; seven slices, two mice; *Figure 2C*, right). Whole-cell recordings of the tagged cells revealed that the majority of intracellularly labeled neurons had axon collaterals located in the hilus (86.7%; 13 HILs and 2 SOMIs with axon in the hilus and inner molecular layer; *Figure 2E*; *Figure 2—figure supplement 1*). None of the labeled cells had axon fibers in the outer molecular layer. Thus, our data indicate that HIL cells form the major anatomical substrate for long-distance DG-septal projections.

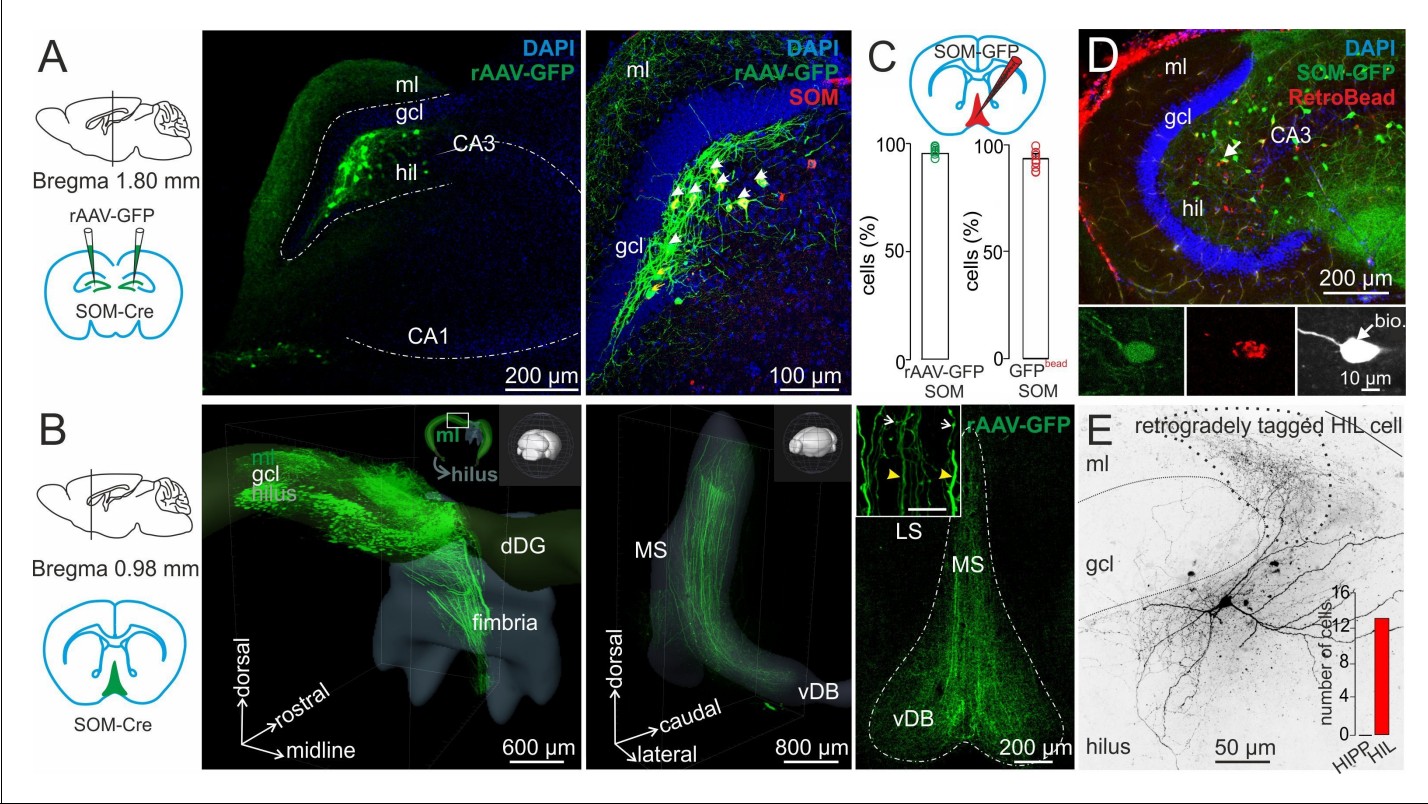

**Figure 2.** HIL cells form long-range projections to the medial septum and vertical diagonal band of Broca (MSvDB). (**A**) *Left*, schematic represents the bilateral injection of rAAV-FLEX-GFP in dorsal DGs of SOM-Cre mice. *Middle*, confocal image stack shows SOM$^+$ somata in the hilus and fiber projections in the molecular layer (ml) and hilus (hil). *Right*, colocalization of immunohistochemically identified SOM (red) and virally expressed GFP. Blue indicates DAPI nuclear staining. *Arrows* point to somata colocalizing SOM and GFP. (**B**) *Left*, schematic illustration of the MSvDB location. *Middle*, three-dimensional Clarity-processed hippocampal SOM-Cre whole mount after rAAV-GFP injection in the dorsal DG. GFP$^+$ projections leave the dorsal DG (dDG) and project toward the fimbria/fornix. *Inset*, illustrates the depicted area of the dorsal DG in relation to the orientation of the entire brain. *Right*, confocal image stack of a frontal 50-μm section of the MSvDB. Dashed line indicates the boarder of the MSvDB. *Inset*, higher magnification of two types of axons in the MSvDB: thick axon with few varicosities and thin axons with several varicosities. (**C–E**) Identification of DG-septal projecting SOMIs. (**C**) Schematic illustrates the injection of the retrograde tracer RedRetroBead (red) in the medial septum. Bar graphs summarize colocalization of SOM (antibody labeling) in the DG of rAAV-GFP injected SOM-Cre mice (*left*) and retrogradely labeled DG cells in SOM-GIN mice (*right*). (**D**) Confocal image stack shows colocalization of GFP in SOM-GIN mice and retrogradely labeled cells (red) after RedRetroBead injection in the MSvDB. *Insets*, arrow points to a RedRetroBead-labelled soma co-expressing SOM-GFP and intracellularly labeled with biocytin (white) during whole-cell recordings. (**E**) Morphological identification of a retrogradely tagged HIL cell. Dotted line indicate area of high axonal density. Continuous line depicts boarders of the granule cell layer (gcl). *Inset*, bar graph summarizes the number of morphologically identified HIL cells projecting to the MSvDB. Bars with lines represent means ± SEM, single circles represent values of individual slices. Abbreviations: gcl, granule cell layer; ml, molecular layer; hil, hilus; dDG, dorsal dentate gyrus; MS, medial septum; LS, lateral septum; vDB vertical limb of the diagonal band of Broca.

The following figure supplement is available for figure 2:

**Figure supplement 1.** Morphological reconstructions of DG-SOMI cells retrogradely labeled from the medial septum.

## Differential excitation of HIPP and HIL cells by inputs from putative granule and mossy cells

How are DG-SOMIs recruited? As previously demonstrated, associative activation of mossy fibers and their target PVIs in the DG leads to a long-lasting increase in the efficacy of glutamatergic transmission and enhanced recruitment of DG-PVIs (*Alle et al., 2001*; *Sambandan et al., 2010*; *Hainmüller et al., 2014*). We therefore asked whether glutamatergic inputs onto DG-SOMIs may also undergo plastic changes upon repetitive associative activation. Due to the hilar location, DG-SOMIs may be targeted by synaptic inputs from GCs (mossy fibers) and mossy cells. We therefore first examined the putative nature of input synapses by positioning an extracellular stimulation pipette at the granule cell layer to

hilus border area and recorded the evoked EPSCs in the two DG-SOMI types in the presence of the GABA$_A$ receptor blocker SR95531 (10 μM; *Figure 3—figure supplement 1*). To distinguish mossy fiber and mossy cell inputs, we bath-applied the group II metabotropic glutamate receptor agonist DCG-IV, which selectively blocks synaptic transmission at mossy fiber synapses (*Sambandan et al., 2010*; *Hainmüller et al., 2014*). EPSCs recorded in HIPPs could be substantially reduced by DCG-IV (by 51.7 ± 3.9%, 10 cells; *Figure 3—figure supplement 1A,B*), indicating their mossy fiber-mediated nature. EPSCs recorded in HIL cells were also diminished by DCG-IV, but to a much lesser mean extent (by 32.6 ± 9.4%, 16 cells; p=0.087, *t*-test; *Figure 3—figure supplement 1B*). More importantly, the magnitude of the DCG-IV effect was highly variable in HIL neurons (range −2.4–96.0%), more variable than in HIPP cells (range 34.1–78.1%; *Figure 3—figure supplement 1B*). Indeed, a substantial proportion of HIL cells showed no or only mild DCG-IV effects (by <20%; 10 out of 16 cells). Thus, our data indicate that both SOMI types are targeted by mossy fibers and HIL cells are preferentially targeted by additional excitatory inputs of different origin, very likely mossy cells (*Larimer and Strowbridge, 2008*).

## Differential forms of synaptic plasticity at glutamatergic synapses targeting HIPP and HIL cells

Next, we applied an associative burst frequency stimulation at 30 Hz (aBFS) to induce plastic changes at glutamatergic SOMI input synapses (*Figure 3*). The aBFS consisted in the presynaptic excitation of glutamatergic inputs with an extracellular stimulation pipette positioned at the granule cell layer to hilus border area paired with timed postsynaptic single action potential induction (see Materials and methods; *Hainmüller et al., 2014*). With this aBFS we aimed to reproduce fast rhythmic neuronal network activity patterns at gamma (30–100 Hz) frequencies observed in the DG of behaving rodents (*Bragin et al., 1995*; *Leutgeb et al., 2007*; *Pernía-Andrade and Jonas, 2014*). Application of the aBFS resulted in a post-tetanic potentiation (PTP) followed by long-lasting depression of synaptically evoked EPSCs in HIPP cells (PTP 176.2 ± 30.6%, mean LTD 15–20 min after aBFS 65.1 ± 17.3% from baseline amplitude; five out of seven recorded HIPP cells showed plastic changes; p=0.01, paired *t*-test; *Figure 3A,B,E*). EPSCs were substantially blocked after LTD induction by DCG-IV indicating their mossy fiber-mediated nature (52.3 ± 7.7%, range 34.1–78.1; five out of five cells; *Figure 3F*). A long-lasting decline in synaptic transmission after plasticity induction outlasted the recording time of 30 min in an additional set of cells indicating that LTD was a stable observation (3 SOMIs; DCG-IV block by 64.7 ± 9.9%; *Figure 3—figure supplement 2*). In marked contrast, the same aBFS resulted in a PTP followed by a long-lasting potentiation of synaptic responses in HIL cells (PTP 203.6 ± 20.4% of baseline amplitude, p<0.001, Mann-Whitney Rank Sum test; mean LTP 153.1 ± 14.9%; 11 out of 12 recorded HIL cells showed plastic changes, p=0.002, paired *t*-test; *Figure 3C,D,E*). Potentiated EPSCs could be reduced by DCG-IV; however, the blocking effect was highly variable among individual HIL cells (by 32.5 ± 9.9%, range −2.4–96.0%, 11 cells; *Figure 3F*) suggesting that LTP was induced at mossy fiber terminals and at other glutamatergic synapses, very likely those originating from mossy cells (*Larimer and Strowbridge, 2008*).

To examine whether plastic changes may depend on the nature of the input synapse, we plotted the magnitude of synaptic plasticity as a function of the DCG-IV blocking effect (*Figure 3F*, right). Both the magnitude of long-lasting depression and the potentiation of synaptic transmission were not correlated with the extent in the blocking effect of DCG-IV on synaptic transmission (Spearman Rank order Correlation, p>0.05 for both comparisons; *Figure 3F*, right). Moreover, the magnitude of synaptic plasticity was not related to changes in the input resistance of the recorded cells. Indeed, the input resistance was stable throughout the entire recording periods (LTD: baseline 326.3 ± 21.2 MΩ *vs* 15–20 min after LTD expression: 323.0 ± 25.5 MΩ, 7 cells; LTP: 201.5 ± 19.2 MΩ *vs* after LTP expression: 204.8 ± 14.6 MΩ, 9 cells; p>0.05, paired *t*-test for both comparisons; *Figure 3—figure supplement 3*). Finally, the sign and magnitude of synaptic plasticity were not related to the peak amplitude of EPSCs obtained during control periods (LTD: baseline EPSC 77.0 ± 19.6 pA 7 cells; LTP: baseline EPSC 74.6 ± 16.4 pA, 12 cells; Mann-Whitney *U* test, p=0.767; Spearman's Rank-Order correlation between the amplitude of EPSCs during baseline and 15–20 min after plasticity induction, p>0.05 for both comparisons).

In summary, long-lasting changes of synaptic transmission are diverse among DG-SOMIs favoring long-lasting depression at HIPP and long-lasting potentiation at HIL cell inputs. These plastic changes seem to neither depend on the intrinsic membrane properties, the initial strength of

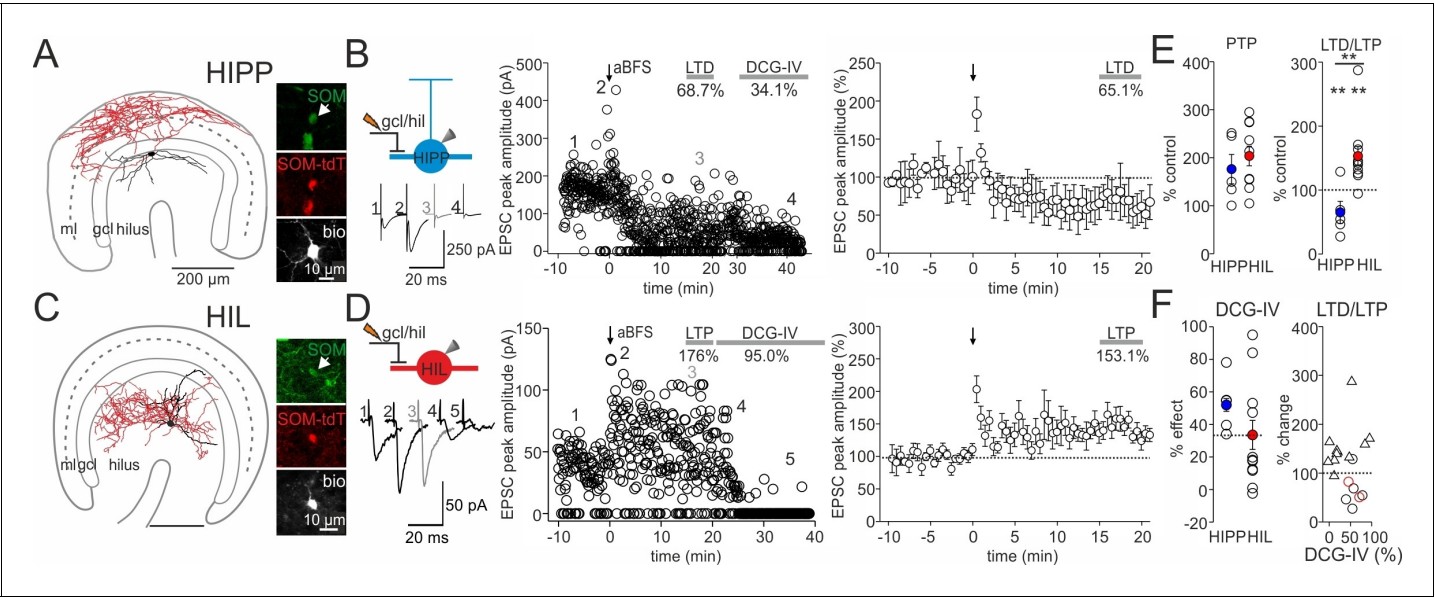

**Figure 3.** DG-SOMI type-specific expression of long-lasting synaptic plasticity. (**A,C**) *Left*, reconstructions of a HIPP and a HIL cell labeled during plasticity experiments shown in B and D. The somata and dendrites are depicted in black and the axon in red. *Right*, confocal images of intrinsic SOM-GFP, antibody labeling against SOM (red) and intracellular biocytin loading (white) of the cells shown on the left. (**B,D**) An associative burst frequency stimulation (aBFS) was applied to glutamatergic input synapses targeting HIPP (**B**) and HIL cells (**D**) to induce long-lasting synaptic plasticity (see Materials and methods). *Left*, schematic illustration of the experimental design. EPSCs were evoked by extracellular stimulation with a pipette positioned at the granule cell layer (gcl) to hilus (hil) border. Individual EPSC peak amplitudes from a single experiment are plotted against time before and after pairing as indicated by the arrow. *Insets* on the *left*, average EPSCs (30 traces) during the baseline period (1), PTP (2), 15–20 min after the induction protocol (3) and after DCG-IV bath-application (4, 5). Time-axes on top (HIPP) was broken between 24 and 28 min. *Right*, summary plot of the time course of EPSC peak amplitudes evoked at glutamatergic HIPP input synapses (five cells). EPSCs were averaged over 30 s intervals and normalized to baseline values. *Note*, the aBFS resulted in a marked long-term depression (LTD). (**D**) *Right*, same as (**C**) for glutamatergic HIL inputs. Application of the aBFS resulted in a PTP followed by a marked long-term potentiation (LTP; 11 cells). (**E**) Summary graphs comparing the magnitude of PTP and LTD/LTP of glutamatergic signals. (**F**) *Left*, graph summarizes the effect of. 1 μM DCG-IV >20 min after plasticity induction on the amplitude of EPSCs in HIPP and HIL cells. *Note*, marked DCG-IV effect in HIPP cells pointing to mossy fiber-mediated nature of synaptic signals but variable DCG-IV effects in HIL cells. *Right*, magnitude of LTD/LTP is not correlated with the DCG-IV effect (Spearman Rank order correlation, p>0.05 for both comparisons). Red circles represent two LTD experiments lacking morphological identification of the recorded SOMIs. Triangles depict HIL and black circles HIPP cells. **p≤0.01. Stars above each group correspond to pairwise comparisons of pre- vs post-aBFS application (paired *t*-test); stars above line correspond to the comparison in plastic changes between HIPP and HIL cells (Mann-Whitney *U* test). Average measurements are represented as mean ± SEM. Circles in E and F depict individual experiments.

The following figure supplements are available for figure 3:

**Figure supplement 1.** DG-SOMIs receive fast glutamatergic synaptic inputs.

**Figure supplement 2.** Robust long-term depression (LTD) at DCG-IV-sensitive inputs onto DG-SOMIs.

**Figure supplement 3.** Synaptic plasticity is independent on changes in the input resistance of recorded SOMIs.

**Figure supplement 4.** Synaptic plasticity is presynaptically expressed.

excitatory input signals nor on the precise origin of the input synapse, but more likely on the nature of the target SOMI.

## Synaptic plasticity at synapses targeting HIPP and HIL cells is presynaptically expressed

To determine the locus of LTD and LTP expression, we examined possible changes in the percentage of transmission failures and performed a coefficient-of-variation (CV) analysis (*Malinow and Tsien, 1990*; *Figure 3—figure supplement 4*). The probability of failures in synaptic signaling

increased by ~99% after LTD (15–20 min after aBFS; from 15.5 ± 5.9% to 30.9 ± 11.3%; 5 HIPP and two non-identified SOMIs; p=0.028, paired *t*-test) but declined by ~74% after LTP induction (from 15.8 ± 6.3% to 7.7 ± 4.2%; 11 HIL cells; p=0.016, Wilcoxon Signed Rank test; *Figure 3—figure supplement 4A*), pointing to its presynaptic origin. Moreover, a plot of the $CV^{-2}$ of the mean EPSC amplitude in the LTD and LTP phase, both normalized to baseline values, revealed that the majority of data points were located close or above the identity line for LTP and close to the identity line for LTD (17 cells in total; *Figure 3—figure supplement 4B*). These data indicate that both forms of plasticity changes are expressed at presynaptic sites.

Taken together, our data provide first evidence for target cell-specific long-lasting changes in synaptic plasticity at DG-SOMIs. LTD and a high rate of transmission failures at HIPP inputs may reduce their recruitment and therefore dendritic inhibition, whereas LTP and a low rate of transmission failures at HIL input synapses will boost their activation and support inhibitory signaling in the hilar DG and the medial septum.

## DG-SOMIs provide local dendritic and perisomatic inhibition onto target cells

The apical dendrites of GCs and fast-spiking interneurons, including PVIs, extend in the outer molecular layer, whereas DG-interneurons form also basal dendrites branching in the deep hilus (*Hosp et al., 2014*). This anatomical organization together with the layer-specific distribution of SOM axons suggests that in addition to dendritic inhibition, DG-SOMIs may also provide perisomatic inhibition. To test this hypothesis, we injected Cre-inducible rAAV vectors encoding ChR2-tdT into the ventral DG of SOM-Cre mice and recorded light-induced IPSCs in postsynaptic GCs and interneurons (*Figure 4*). 1-Photon laser pulses (0.5 ms, 473 nm) were applied either to the outer molecular layer or close to the soma of the recorded cell to evoke distal vs perisomatic inhibitory signals, respectively ($IPSC_{oml}$; $IPSC_{psoma}$; in total: 12 GCs, four fast-spiking and two regular-spiking interneurons; three simultaneous recordings of an interneuron and a GC in the same slice; *Figure 4B*). IPSCs recorded in GCs had always a small peak amplitude and slow rise time independent on the light-pulse location (amplitude $IPSC_{oml}$ 9.6 ± 1.2 pA vs $IPSC_{psoma}$ 8.7 ± 1.6 pA, p=0.4327; *Figure 4C*; rise time $IPSC_{oml}$ 1.8 ± 0.2 ms vs $IPSC_{psoma}$ 2.5 ± 0.6 ms; p=0.2094, two-tailed Wilcoxon Rank Sum test) suggesting their distal dendritic origin. On the contrary, distally induced IPSCs in interneurons, were 6.9-fold smaller than the ones induced close to the soma ($IPSC_{oml}$ 23.0 ± 6.4 pA vs $IPSC_{psoma}$ 159.6 ± 72.5 pA, p=0.028, paired Wilcoxon Rank Sum test; *Figure 4B,C*) and had a significantly slower rise time ($IPSC_{oml}$ 1.8 ± 0.6 ms vs $IPSC_{psoma}$ 0.7 ± 0.2 ms; p=0.046, paired Wilcoxon Rank Sum test), suggesting that in addition to dendritic inhibition, interneurons receive powerful perisomatic SOMI-mediated synaptic inhibition. Indeed, high-resolution confocal images revealed $SOM^+$ boutons at the soma of PVIs (13 cells; *Figure 4A*, inset). Consistent with the proposed synapse location, somatically evoked IPSCs in interneurons had a 3.4-fold faster rise time and an 18.3-fold larger amplitude than the ones evoked in GCs (p=0.0037 and p=0.00074, respectively, Wilcoxon Rank Sum test; *Figure 4D*). This is further reflected in a ~sixfold higher amplitude ratio between $IPSC_{psoma}$ and $IPSC_{oml}$ in interneurons than in GCs (5.8 ± 1.3 *vs* 1.0 ± 0.2; p=0.000078, *t*-test; *Figure 4D*). Thus, HIPP cells provide dendritic inhibition onto GCs and interneurons, whereas $SOM^+$ axons in the hilus seem to supply powerful perisomatic inhibition onto interneurons (*Figure 4E*).

To further prove that HILs are synaptically connected to interneurons in the DG, we performed paired HIL-interneuron recordings and examined their functional properties (*Figure 4F,G*). Single action potentials in presynaptic HILs evoked unitary IPSCs (uIPSCs) in target interneurons (two basket cells, two HILs; *Figure 4F*) after a short latency (1.6 ± 0.25 ms), with a fast rise time (0.6 ± 0.06 ms) and a fast decay time constant (τ = 11.2 ± 0.6 ms; *Figure 4G*). The amplitude was variable with a mean value of 59.1 ± 17.6 pA (range 23.4–91.8 pA). Thus, HILs provide perisomatic inhibition to DG-interneurons including basket cells and HILs.

## DG-SOMIs form synaptic contacts onto septal GABAergic, cholinergic and glutamatergic cells

Which cell types in the medial septum are targeted by HIL cells? To address this question, we combined rAAV-FLEX-GFP injections in the dorsal DG of SOM-Cre mice to label HIL cells projecting to

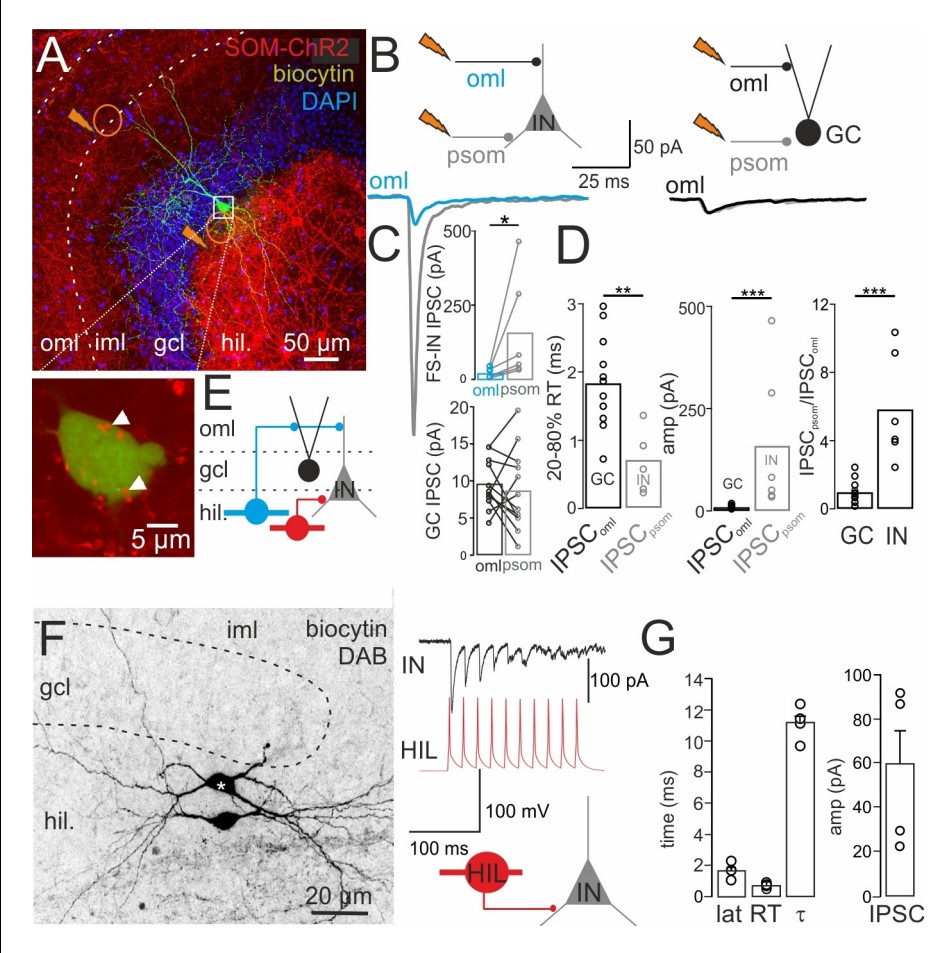

**Figure 4.** DG-SOMIs provide dendritic and perisomatic inhibition onto DG target cells. (**A**) *Upper*, confocal image stack shows expression of channelrhodopsin-2 (ChR2)-tdTomato (tdT) in SOM[+] interneurons upon stereotaxic injection of rAAV-ChR2-tdT in the ventral DG (Materials and methods). Orange circles with jags indicate location of 1-Photon light pulses (0.5 ms, 473 nm) applied to the outer molecular layer (oml) and at the perisomatic (psom) area in an alternating manner to evoke IPSCs in GCs and fast-spiking interneurons (INs). A representative IN with axon arborizations restricted to the granule cell layer (gcl), identifying it as basket cell, was labeled intracellularly during whole-cell recordings and is shown in green. The magnified soma of the same cell is shown below. Arrow heads point to SOM[+] bouton-like varicosities. (**B**) IPSCs recorded in an IN (left) and a GC (right) upon light-pulse application to the oml (blue and black traces, respectively) are superimposed with IPSCs evoked close to the soma (psom; gray traces). (**C**) Bar graphs summarize amplitude (amp) of oml and psom evoked IPSCs in INs and GCs. (**D**) Summary of the 20–80% rise time (RT), peak amplitude (amp) and IPSC$_{psom}$/IPSC$_{oml}$ ratio of IPSCs recorded in GCs and INs (12 GCs, 6 INs including four fast-spiking and two regular-spiking INs, 3 GCs and three fast-spiking INs were recorded simultaneously in one slice). *Note*, perisomatically evoked IPSCs in INs are larger and faster than the ones evoked by light pulses applied to the oml in GCs. (**E**) Schematic illustrates connections among DG-INs (blue HIPP, red HIL cell) and GCs (black filled circle). (**F**) *Left*, intracellular labeling of a synaptically connected HIL-HIL pair. Star indicates the presynaptic cell. *Right*, a train of 10 action potentials (50 Hz) in the presynapstic HIL (red) evoked unitary IPSCs (uIPSCs) in the postsynaptic IN (black; traces correspond to the pair shown on the left). (**G**) Summary plots showing the functional properties of uIPSCs from four HIL-IN pairs (two HIL-HIL, two HIL-basket cells). Circles represent single data points. Circles connected by lines correspond to one experiment. Bars with lines indicate means ± SEM; *p≤0.05, **p≤0.01, ***p≤0.001. Abbreviations: lat, synaptic latency; RT, 20–80% rise time and τ, decay time constant of uIPSC; oml, outer molecular layer; psom, perisomatic.

the MSvDB with antibody labeling against the two neurochemically defined main neuron classes in the medial septum, PVIs and cells expressing choline acetyltransferase (ChATs; *Figure 5*). Glutamatergic cells as the third important neuronal element of the MSvDB were putatively identified during whole-cell recordings on the basis of their distinct discharge patterns, characterized by bursts of action potentials interleaved by silent periods (*Figure 5D*, inset; *Figure 6C*; *Manseau et al., 2005*). By using confocal microscopy, we identified putative synaptic contacts, defined as presynaptic SOM-GFP$^+$ axon varicosities co-localizing gephyrin, a postsynaptic marker of GABAergic terminals (*Alldred et al., 2005*; *Fritschy et al., 2008*), in close proximity to cell bodies or proximal dendrites of cholinergic cells and PVIs (67 ChAT neurons tested in six slices, three mice; 126 PVIs tested in 13 slices; three mice; *Figure 5A–C*). In case of putative glutamatergic cells as defined by their discharge activity (*Figure 5D*, inset), we observed SOM$^+$ axonal fibers in close vicinity of intracellularly labeled

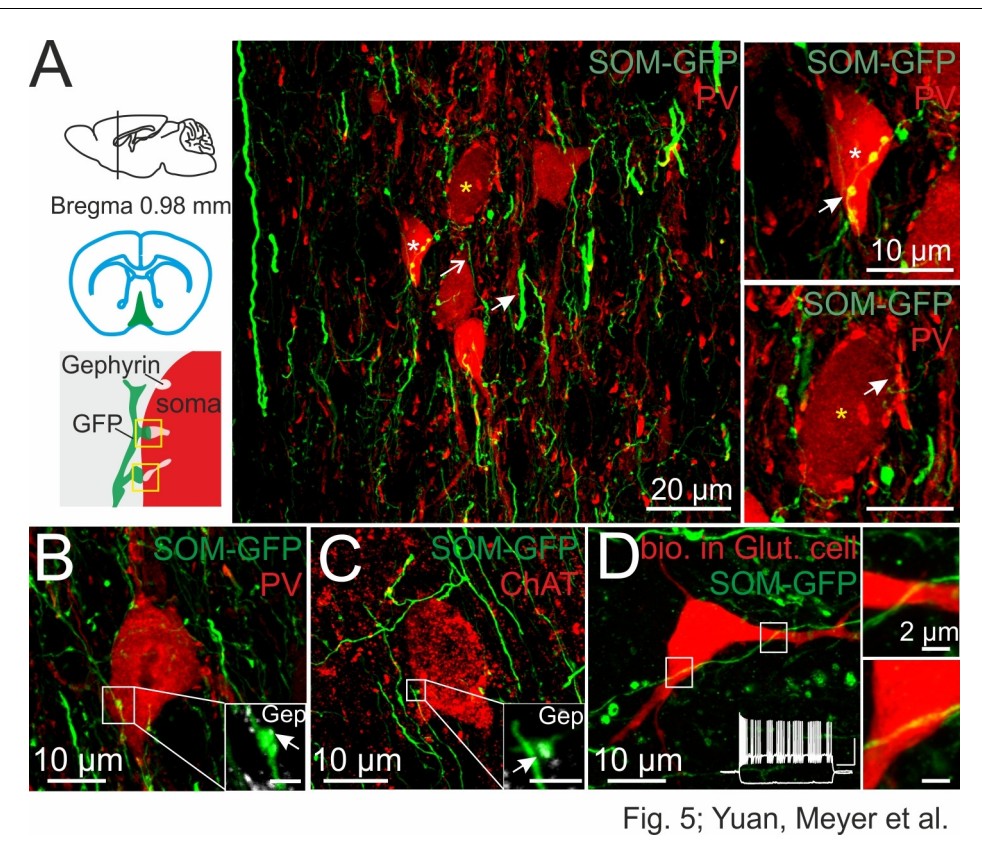

**Figure 5.** DG-SOMIs form putative synapses onto septal GABAergic, cholinergic and putative glutamatergic cells. (**A**) Confocal image stack of SOM fibers expressing GFP in the medial septum and vertical limb of the diagonal band of Broca (MSvDB) upon rAAV-FLEX-GFP injection bilaterally in the dorsal DG of SOM-Cre mice. Thin GFP-positive fibers (open arrow) form 'en passant' bouton-like varicosities at close proximity to cell bodies of PVIs (red). Somata marked with a white and yellow star are shown on the right at higher magnification. *Top right*, arrow points to putative synaptic contacts formed by DG-septal SOMIs. *Bottom right*, PVI cell bodies are surrounded by PV-expressing boutons very likely originating from local PVIs (arrow). (**B,C**) Confocal image stacks of putative synaptic contacts formed by DG-septal projecting SOMIs at a PVI soma (**B**) and at a cell body expressing choline acetyltransferase (ChAT, (**C**). *Insets*, high magnifications of the putative contact sites colocalizing gephyrin (Gep; arrow). Scale bar, 2 μm. (**D**) Intracellularly labeled cell in the MSvDB with biocytin during whole-cell recordings (red). The cell showed a burst-like discharge pattern (inset; cluster-firing cells) during depolarizing current injections (1 s, 300 pA; −100 pA) characteristic for glutamatergic (Glut) neurons in the MSvDB (*Manseau et al., 2005*; *Mattis et al., 2014*). White boxes are magnified on the right and show SOMI-GFP fibres in close proximity of the soma and the proximal dendrite of the putative glutamatergic cell. *Note*, lacking bouton-like shape of this putative contact site.

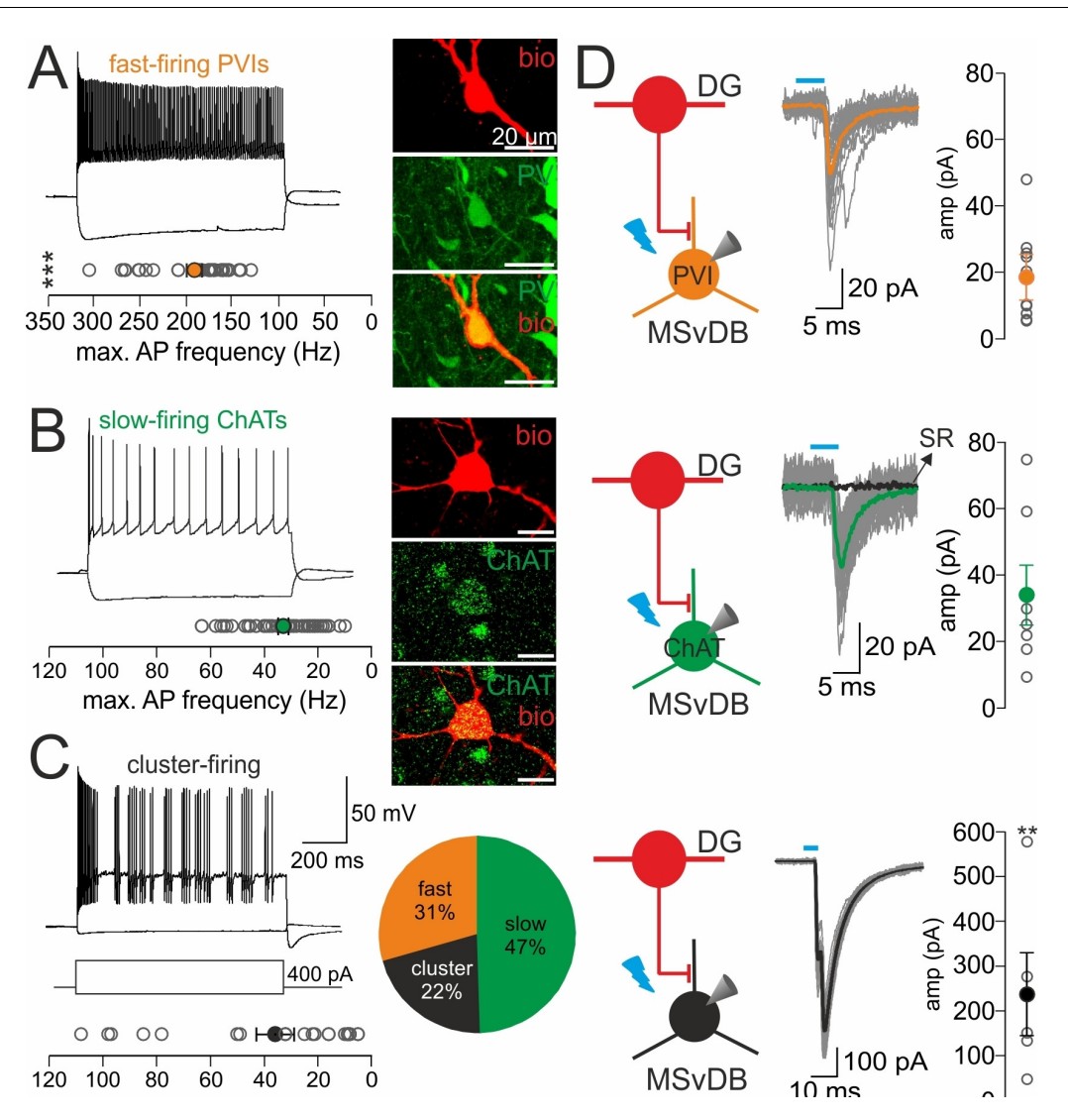

**Figure 6.** DG-SOMIs provide weak inhibition on GABAergic and cholinergic but strong inhibition onto putative glutamatergic cells in the medial septum. (**A–C**) Passive and active membrane properties and the neurochemical marker contents of the three main neuron types in the medial septum and vertical limb of the diagonal band of Broca (MSvDB). *Left*, characteristic discharge patterns of the three cells types (1 s, −100 and 400 pA current injection) classify them as fast-spiking (**A**), slow-firing (**B**) and cluster-firing (**C**) cells. Plots summarize the maximal discharge frequency of the three cell types. The discharge frequency between PVIs and slow-firing as well as cluster-firing cells was significantly different (***p<0.001 for both comparisons, *t*-test). *Right*, intracellular labeling of cells with biocytin (red) with subsequent antibody labeling (green). *Note*, all fast-firing cells expressed parvalbumin (PV). Slow-firing cells express choline acetyltransferase (ChAT) and cluster-firing cells have been previously identified as glutamatergic (Glut) cells (*Manseau et al., 2005*; *Mattis et al., 2014*). Pie chart summarizes the relative proportion of the recorded cell types (32 fast-spiking PVIs, 49 slow-firing ChATs, 23 cluster-firing putative Glut cells). (**D**) *Left*, schematic illustration of the experimental procedure. DG-septal projecting SOMIs expressing channelrhodopsin-2 (ChR2) after injection of rAAV-FLEX-ChR2-tdT in the dorsal DG were activated by light-pulses (5 ms, full-field illumination, 473 nm) applied to the MSvDB (red circles represent HIL cells). *Middle*, IPSCs recorded in the three neuron types. Individual IPSCs (grey traces) and average IPSCs (color-coded traces) are shown superimposed. Bath application of 10 µM SR59931 blocked IPSCs in three slow-firing ChAT cells. *Right*, summary plots show peak amplitudes of evoked IPSCs in PVIs (10 out of 32 cells), ChATs (7 out of 49 cells) and putative glutamatergic cells (5 out of 23 cells). Open circles are individual data points, filled circles are mean values with lines representing ± SEM; **p≤0.01, Mann-Whitney *U* test for pair-wise comparisons between PVIs vs putative glutamatergic cells and ChATs vs glutamatergic cells. ***p<0.001 for pair-wise

*Figure 6 continued on next page*

*Figure 6 continued*

comparisons between PVIs vs putative glutamatergic cells and PVIs *vs* ChATs. To compare three data sets for significant differences a Kruskal-Wallis one-way analysis of variance on Ranks was performed.

The following figure supplement is available for figure 6:

**Figure supplement 1.** Sequential recordings of fast-spiking PVIs and cluster-firing putative glutamatergic cells in the same slice preparation.

cell bodies or proximal dendrites. However, they lacked a bouton-like shape as well as postsynaptic gephyrin expression (23 cells tested; *Figure 5D*).

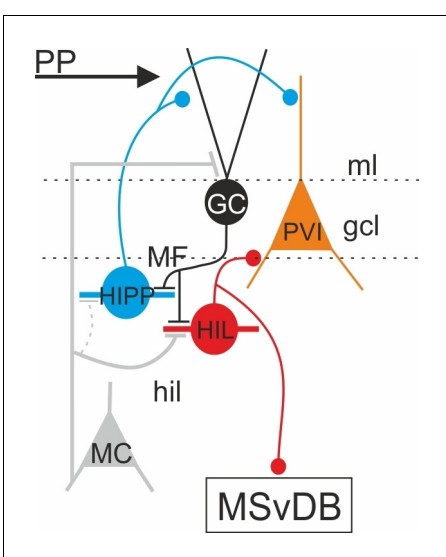

**Figure 7.** Schematic of the dentate gyrus neuronal network with some of the main cellular components. Schematic illustration of the synaptic integration of DG-SOMIs in the local dentate gyrus (DG) and the medial septum and vertical limb of the diagonal band of Broca (MSvDB) circuitry. The perforant path (PP) transmits information from the entorhinal cortex to the DG by targeting distal dendrites of granule cells (GCs) and GABAergic cells including PVIs (orange). DG-SOMIs consist of at least two contrasting types. HIPP (blue) and HIL (red) cells are recruited by GC inputs via mossy fiber (MF) synapses (black lines with bars) and glutamatergic inputs from mossy cells (MCs) which show target preference for HIL cells (grey lines with bars). Repetitive associative activation of glutamatergic inputs induces long-lasting depression of synaptic transmission onto HIPP cells but long-lasting potentiation onto HIL neurons. HIPPs provide weak and slow dendritic inhibition onto local GCs and interneurons, including PVIs. HILs provide perisomatic inhibition onto local DG-interneurons including PVIs and additionally form extra-DG long-range projections to the medial septum to strongly inhibit cluster-firing putative glutamatergic cells and to mildly inhibit fast-spiking PVIs and slow-discharging cholinergic cells.

## DG-SOMIs provide strong inhibition onto putative septal glutamatergic but weak inhibition onto GABAergic and cholinergic cells

To test whether morphologically identified synaptic contacts in the MSvDB are functional, we injected rAAVs-FLEX-ChR2-tdT bilaterally in the dorsal DG of SOM-Cre mice. Two weeks after viral expression, slices of the MSvDB were prepared to record IPSCs in cells of the MSvDB evoked by full-field illumination (Materials and methods; *Figure 6*). Cells were identified during recordings on the basis of their characteristic electrophysiological properties (*Markram and Segal, 1990*; *Morris et al., 1999*; *Manseau et al., 2005*) and in case of fast- and slow-firing cells on the basis of their PV and ChAT expression, respectively. Fast-spiking PVIs formed 31% of the recorded neuron population and discharged action potentials with short half-duration and high maximal frequency (half duration: $0.43 \pm 0.02$ ms; frequency: $191.2 \pm 8.1$ Hz; adaptation ratio: $1.8 \pm 0.1$, input resistance: $318.5 \pm 40.3$ MΩ, 32 cells; *Figure 6A*). ChATs formed with 47% the majority of recorded cells (49 cells; *Figure 6B*) and discharged action potentials with a significantly broader half-width, lower discharge frequency and larger adaptation of spike trains than PVIs (half duration: $0.9 \pm 0.03$ ms; frequency: $32.8 \pm 1.9$ Hz; adaptation ratio: of $5.3 \pm 0.4$; $p<0.0001$ for all three pairwise comparisons; *t*-test; *Figure 6B*) consistent with previous reports (*Markram and Segal, 1990*). The remaining 22% of the cells generated bursts of action potentials characteristic for glutamatergic cells (half duration: $0.8 \pm 0.06$ ms; frequency: $35.8 \pm 7.0$ Hz; input resistance: $387.9 \pm 42.7$ MΩ; 23 cells; *Manseau et al., 2005*; *Huh et al. 2010*; *Figure 6C*). Thus, the main neuron types in the MSvDB could be unequivocally identified on the basis of their electrophysiological characteristics or their neurochemical marker content (*Figure 6A,B*, right).

To directly compare the strength of DG-SOMI-mediated synaptic signals among the three neuron types, we systematically increased the light intensity and recorded maximally evoked IPSCs. Synaptic signals with the largest mean peak amplitude were obtained from putative glutamatergic neurons (237.0 ± 92.6 pA; 5 out of 23 tested cells; *Figure 6D*, bottom), whereas ~12.8 and ~7.0 times smaller IPSCs were recorded from PVIs and ChATs, respectively (PVI 18.5 ± 6.9 pA, 10 out of 32 tested cells; ChAT 33.8 ± 9.0 pA, 7 out of 49 tested cells; p<0.01 for both comparisons, Mann Whitney *U* test; *Figure 6D*). IPSCs could be blocked by 10 µM SR95531 indicating their GABA$_A$ receptor-mediated nature (98.3 ± 4.2% block, 3 ChAT cells tested; *Figure 6D*). In a subset of experiments two cell types were subsequently recorded in the same slice at identical illumination conditions to confirm the differential strength of inhibition (three PVI and cluster-firing cells; three PVI and slow-firing ChAT cells; *Figure 6—figure supplement 1*). Thus, DG-septal projecting SOMIs provide strong inhibition onto putative glutamatergic neurons but mild inhibition onto slow-firing ChAT cells and fast-spiking PVIs.

## Discussion

For a long time, HIPP cells have been considered to be synonymous to DG-SOMIs (*Freund and Buzsáki, 1996*; *Mott et al., 1997*). Here, we provide first evidence that DG-SOMIs are diverse and divide at least into two functionally contrasting types on the basis of their morphological characteristics, their intrinsic membrane properties, the nature of their excitatory inputs and postsynaptic target specificity. Their functional embedding into the DG circuitry allows both SOMI types to contribute to the processing of spatial information transmitted by the entorhinal cortex in a highly cell-type-specific manner (*Figure 7*). The majority of DG-SOMIs studied here are not directly recruited by the perforant path-mediated excitatory drive but indirectly through GCs. HIPPs receive fast GC-mediated excitatory inputs and provide lateral feedback dendritic inhibition onto large DG neuronal populations. They are therefore prompted to control size and stability of GC assemblies encoding spatial information (*Stefanelli et al., 2016*). On the contrary, HILs are activated by GCs and other glutamatergic cells, very likely mossy cells, supplying fast and strong perisomatic inhibition onto local interneurons including PVIs and SOMIs. Evidence for SOMI-mediated inhibition onto local mossy cells are so far lacking (*Deller and Leranth, 1990*; *Acsády et al., 2000*), but cannot be fully excluded (*Larimer and Strowbridge, 2008*). HIL cells form a functional link to the medial septum and could thereby be involved in the coordination of local processing of spatial and contextual information provided by the perforant path with theta oscillations driven by the medial septum. This proposal fits to single-cell recordings of hippocampal-septal projecting GABAergic cells which generate action potentials phase locked to theta cycles recorded in the hippocampus and the medial septum (*Losonczy et al., 2002*; *Jinno, 2009*).

### Functional interaction of the DG with other brain areas via SOMI projections

The DG, similar to other cortical regions, shows a distinct laminar structure with layer-specific distribution of afferent pathways and local axon collaterals (*Ramón, 1968*; *Amaral and Witter, 1989*; *Förster et al., 2006*). The main extra-hippocampal afferent projection to the hippocampus, the perforant path, originates in layers II and III of the entorhinal cortex and terminates on the distal apical dendrites of GCs and interneurons (*Ramón, 1968*; *Amaral and Witter, 1989*; *Witter, 2007*; *Figure 7*). This pathway provides a rich multimodal stream to the DG, including processed sensory information from neocortical areas. Other afferent pathways comprise inter-areal connections, such as the commissural-associational fibers from hilar mossy cells to the ipsi- and contralateral DG, terminating on proximal dendrites of GCs and interneurons in the inner molecular layer (*Scharfman, 2016*). Local GC axons project to the hilus of the DG and to the CA3 and target with their mossy fiber synapses the dendrites of CA3 principal cells. Long-range inhibitory projections also follow these pathways to interconnect the DG with other cortical areas. First, GABAergic long-distance bi-directional inhibitory routes have been morphologically identified between the entorhinal cortex and the DG (*Jinno, 2009*; *Melzer et al., 2012*; *Caputi et al., 2013*). Second, retrograde tracing in combination with immunohistochemistry revealed that hilar SOMIs project to the contralateral DG (*Leranth et al., 1990*). Third, in vitro intracellular labeling of DG interneurons with cell bodies in the molecular layer showed that axons of these cells target the subiculum by crossing the hippocampal fissure (*Ceranik et al., 1997*).

Thus, previous investigations together with the data presented here show that the DG is strongly inter-connected with other cortical areas via GABAergic routes.

In contrast to the limited information on GABAergic projections from the DG to the medial septum (*Takács et al., 2008*), multiple hippocampal-septal inhibitory connections have been identified (*Jinno, 2009*; *Caputi et al., 2013*). They are largely formed by SOMIs located in the stratum oriens of CA1 and CA3 and stratum lucidum of CA3 (*Alonso and Köhler, 1982*; *Tóth and Freund, 1992*; *Jinno and Kosaka, 2002*; *Gulyás et al., 2003*; *Ferraguti et al., 2005*; *Takács et al., 2008*). By comparing the morphology of these hippocampus-septal cell types with the ones identified in this study, we revealed some similarities. Both hippocampal- and DG-septal cells form local in addition to long-range septal synaptic contacts (*Jinno et al., 2007*; *Figures 1* and *2*) and target, cholinergic, GABAergic and putative glutamatergic cells (*Tóth et al., 1993*; *Gulyás et al., 2003*; *Jinno et al., 2007*; *Figures 5* and *6*). Hippocampal-septal cells have been shown to form the highest number of morphologically identified terminals onto GABAergic and the lowest on cholinergic cells (*Tóth et al., 1993*). These observations are in apparent contrast to our electrophysiological data showing that DG-septal cells evoke strong inhibitory signals in putative glutamatergic neurons and weak signals in GABAergic and cholinergic cells (*Figure 6*). These differences may be explained by disparities in the convergence of brain-area-specific SOM$^+$ fibers projecting to the medial septum or different release probabilities of their output synapses.

## Functional relevance of DG-SOMIs in local information processing

The present results provide an anatomical and physiological solution to the question of how theta oscillations in the DG could be coordinated with the medial septum. In the classical view, cholinergic inputs from the medial septum are critical for the emergence of theta oscillations in the hippocampus (*Stewart and Fox, 1990*) and the DG (*Pabst et al., 2016*). However, a further key component is the rhythmically active septal inhibitory drive as revealed by simultaneous local field potential recordings in CA1 and the medial septum (*Hangya et al., 2009*). Indeed, selective deletion of septal GABAergic cells resulted in a marked reduction of hippocampal theta power (*Yoder and Pang, 2005*) and spatial memory (*Pang et al., 2011*). These GABAergic cells contact exclusively interneurons in the hippocampus (*Gulyás et al., 1990*; *Borhegyi et al., 2004*; *Hangya et al., 2009*; *Hassani et al., 2009*). The major GABAergic route from the medial septum to the hippocampus is mediated by PVIs (*Köhler et al., 1984*; *Tóth et al., 1993*). They have been shown to target almost exclusively GABAergic cells in CA1 and CA3 (*Leranth et al., 1990*; *Unal et al., 2015*). Most of these targets have been demonstrated to be the source of back-projections to the medial septum (*Tóth et al., 1993*; *Takács et al., 2008*) thereby forming a bi-directional hippocampal-septal inter-neuron loop. We propose that a bi-directional interneuron loop may also exist between the DG and the medial septum. Indeed, virtually all DG-septal cells are SOMIs and contact all main neuron types in the MSvDB (PVIs, ChATs, putative glutamatergic cells; *Figures 5* and *6*). Our unpublished observations suggest that septal PVIs in turn target somata and proximal dendrites of DG-PVIs and – SOMIs (data not shown). In addition to the DG-septal loop proposed here, long-distance interneuron-interneuron interactions have been demonstrated between cortical and subcortical areas in mammals and humans (*Linkenkaer-Hansen et al., 2005*; *Guitart-Masip et al., 2013*) suggesting that GABAergic loops may be a common principle contributing to the dynamic coupling of brain areas for conjoint processing of information and control of behavior.

A further important theta-modulated excitatory drive is provided to the DG from the entorhinal cortex via the perforant path (*Bragin et al., 1995*). Embedded in the DG, local interneurons receive excitatory inputs from the perforant path depending on their dendritic distribution (*Bartos et al., 2011*). Therefore, the majority of DG-SOMIs are not directly recruited by the perforant path but indirectly by GCs (*Figure 7*). During repetitive activation of the perforant path at theta frequencies, long-lasting plasticity at mossy fiber synapses may add a new level of functional integration of interneurons in the DG circuitry. De-potentiation of mossy fiber inputs onto HIPP cells emerges postsynaptic to GCs which have been repeatedly activated by the perforant path and thus have been themselves subject to potentiation (*Schmidt-Hieber et al., 2004*). As a consequence of long-lasting depression of synaptic transmission at mossy fiber terminals, HIPP cells will be less recruited, reduce their dendritic inhibition onto strongly activated GCs and thereby support long-lasting strengthening of perforant path-mediated synaptic signals onto GC dendrites (*Miles et al., 1996*). On the contrary, the same strongly recruited GC population will strengthen their synaptic inputs onto PVIs. This will

enhance their recruitment and thereby will boost perisomatic inhibition onto GC populations (*Sambandan et al., 2010*; *Hainmüller et al., 2014*). As a consequence, functional associations of few sparsely active GCs and interneurons will emerge. The proposed reduced recruitment of HIPPs might support the flow of information from the entorhinal cortex to the DG, whereas enhanced activation of PVIs will improve the signal-to-noise ratio during cell assembly formation, promote segregation of information and thereby enhance the storage capacity of the network (*Strüber et al., 2015*). Potentiation of mossy fiber inputs of the same strongly active GC population and mossy cells onto HIL neurons will boost their recruitment and thereby support the temporal coordination of activity patterns from local cell assemblies with the ones generated in the septum. Consistent with this theory, optogenetic silencing of DG-SOMIs resulted in the loss of GC engrams and spatial context recognition (*Stefanelli et al., 2016*).

In summary, the functional diversity of SOMIs adds a new dimension to the complex functionality of DG neuronal networks. Dendritic-inhibition provided by HIPP cells will control the flow of spatial information from the entorhinal cortex to the DG, whereas perisomatic inhibition mediated by HIL cells will support the temporal coordination of local rhythmic DG activity with the activity patterns governed by the medial septum. Such coordination could be of particular importance during navigation when spatial information processing is temporally coordinated with running speed (*Fuhrmann et al., 2015*).

## Material and methods

### Electrophysiology

Transverse hippocampal slices (300 µm) were cut with a VT 1200 s vibratome (Leica, Germany) from 18- to 35-days-old transgenic mice expressing green fluorescent protein (GFP) in SOM-expressing inhibitory interneurons (GIN mice; mice homozygous for the TgN(GadGFP)45704Swn transgene express Enhanced Green Fluorescent Protein [EGFP] under the control of the mouse *Gad1* [GAD67] gene promoter; *Oliva et al., 2000*) or SOM-Cre mice (SOM-IRES-Cre; Cre recombinase is expressed under the control of the endogenous *Sst* promoter; Jackson Laboratories, Stock no. 003718) crossed with Ai9-RCL-tdT reporter mice hemizygous for Rosa-CAG-LSL-tdTomato-WPRE (SOM-tdT; Jackson Laboratories, Stock no. 007909). GIN mice were used for interneuron identification (*Figure 1*). GIN as well as SOM-tdT animals were used for the examination of plastic changes at SOMI input synapses (15 GIN and 6 SOM-tdT mice; *Figure 3*, *Figure 3—figure supplement 2*). All animal procedures were performed in accordance to national and institutional legislations (license no.: G-11/53; X-12/20D).

Acute hippocampal slices were perfused with an artificial cerebrospinal fluid (ACSF) consisting of (in mM) NaCl 125, NaHCO$_3$ 25, KCl 2.5, NaH$_2$PO$_4$ 1.25, D-glucose 25, CaCl$_2$ 2, MgCl$_2$ 1 (equilibrated with 95% O$_2$/5% CO$_2$,) for 20–30 min (29–34°C) and then stored at room temperature (22–24°C). Recording pipettes (wall thickness: 0.5 mm; inner diameter: 1 mm) were pulled from borosilicate glass tubing (Hilgenberg, Germany; Flaming-Brown P-97 puller, Sutter Instruments, USA), filled with a solution containing (in mM) K-Gluconate 110, KCl 40, HEPES 10, MgCl$_2$ 2, Na$_2$ATP 2, EGTA 0.1% and 0.2% biocytin (Molecular Probes) and in some experiments Alexa Fluor 488 (150 µM) (pH = 7.2; 290–310 mOsm), resulting in a final pipette resistance of 3–6 MΩ. The internal solution for measurements of synaptic plasticity contained in mM: K-gluconate 120, KCl 20, EGTA 0.1, MgCl$_2$ 2, Na$_2$ATP 4, GTP 0.5, HEPES 10, Na$_2$-phosphocreatine 7, spermine terahydrochloride 0.1% and 0.2% biocytin (pH = 7.2). In a subset of three experiments for synaptic plasticity, we applied pipettes with a resistance >10 MΩ to reduce the possibility of wash-out of internal components (*Figure 3—figure supplement 2*). GFP- or tdT-expressing interneurons were identified during the experiment using epifluorescence illumination. Recordings were obtained from neurons in the DG under visual control using infrared differential interference contrast video microscopy (*Sauer and Bartos, 2010*). To evoke synaptic excitatory signals, we positioned an extracellular monopolar stimulation pipette made of glass capillaries and filled with a sodium-rich, HEPES-buffered solution containing in mM: 135 NaCl, 5.4 KCl, 1.8 CaCl$_2$, 1 MgCl$_2$ and 5 HEPES (*Sambandan et al., 2010*), at the granule cell layer to hilus boarder. Excitatory postsynaptic currents (EPSCs) were evoked by short depolarizing voltage pulses (0.1–0.2 ms; 5–10 V) in the presence of 5–10 µM 4-[6-imino-3-(4-methoxyphenyl)pyridazin-1-yl]butanoic acid hydrobromide (SR95531) added to the extracellular solution. GC-mediated inputs were

identified on the basis of their fast time course and sensitivity to bath-applied 1 µM (2S, 2R, 3R)−2-(2, 3-dicarboxycyclopropyl) glycine (DCG-IV; *Sambandan et al., 2010*; *Hainmüller et al., 2014*).

During paired whole-cell patch clamp recordings of synaptically connected interneurons, single action potentials were evoked by brief depolarizing current injection in the presynaptic interneuron (1–2 ms, 0.4–1.0 nA) and unitary inhibitory postsynaptic currents (uIPSCs) were recorded at −70 mV holding potential in the postsynaptic cell. Paired recordings were performed in the presence of 20 µM 6-cyano-7-nitroquinoxaline-2, 3-dione (CNQX; Sigma-Aldrich, USA) to block EPSCs. All recordings were performed with one Multiclamp 700B amplifier (Molecular Devices, USA). Series resistance ($R_s$; 15–20 MΩ) was compensated in voltage-clamp at 75–85% (20–30 µs time lag) and in current-clamp at 100% (5–10 µs time lag) during single whole-cell recordings for the identification of intrinsic membrane properties and during paired recordings. $R_s$ was not compensated during septal opto-physiological and long-term synaptic plasticity experiments but continuously monitored by applying 10 mV test-pulses. Signals were filtered at 5–10 kHz and digitized at 20–40 kHz with a Power1401 laboratory interface (Cambridge Electronic Design, UK). Stimulus-generation and data acquisition were performed with a custom-made Igor-based program (FPulse, Dr. Fröbe, Institute for Physiology I, University Freiburg; available at: http://www.physiologie.uni-freiburg.de/research-groups/neural-networks). Recording temperature was 31–34°C.

For the induction of synaptic plasticity, we followed our previously applied associative plasticity induction protocol (aBFS; *Alle et al., 2001*; *Sambandan et al., 2010*; *Hainmüller et al., 2014*). In brief, extracellular stimulation of synaptic inputs at 30 Hz bursts of 25 pulses, repeated 12 times every 3 s, was paired with action potential generation in the postsynaptic SOMI. Action potentials were evoked by 1 ms-long depolarizing current injections with a 3 ms delay following the peak of the synaptically evoked signal (holding potential of −70 mV). In some experiments, extracellular stimulation was strong enough to induce a postsynaptic action potential. In these cases, additional depolarizing pulses in the postsynaptic cell were not applied. PTP was determined from peak amplitudes of EPSCs evoked 0–30 s after plasticity induction. Synaptic plasticity was obtained from EPSC amplitudes recorded 15–20 min after the aBFS (*Figure 3*) and in longer lasting experiments during 28 to 32 min after the BFS (*Figure 3—figure supplement 2*). EPSC amplitudes were normalized to the mean EPSC recorded during baseline periods preceding the induction protocol. The membrane resting potential did not change by ±4 mV throughout experiments. Data were usually discarded if the $R_s$ changed more than 25% except of three cases in which $R_s$ changes > 25% did not preclude the expression of strong LTP. Mean EPSC values include failures. Data were not corrected for baseline noise.

## Optophysiology

For cell-type- and brain-area-specific excitation of SOMIs, we used recombinant adeno-associated viruses (rAAVs) encoding the light-sensitive channelrhodopsin-2 (ChR2) and the red fluorophore tdT. The virus (rAAV-FLEX-ChR2-tdT) was produced from plasmid pCAG-FLEX-ChR2-tdT (gift from Dr. Scott Sternson) as described previously (*McClure et al., 2011*). Briefly, virions containing a 1:1 ratio type one and type two capsid proteins were produced by transfecting human embryonic kidney (HEK) 293 cells with pCAG-FLEX-ChR2-tdT and AAV1 (pH21), AAV2 (pRV1) helper plasmids plus the adenovirus helper plasmid pFdelta6 using the calcium phosphate method. 48 hr after transfection cells were harvested and rAAV-FLEX-ChR2-tdT was purified using 1 ml HiTrap heparin columns (Sigma) and concentrated using Amicon Ultra centrifugal filter devices (Millipore). Infectious particles (viral titer) were calculated by transducing Cre-recombinase expressing HEK293 cells and counting tdT$^+$ cells.

The rAAV-FLEX-ChR2-tdT was stereotaxially injected bilaterally into the dorsal (4 µl of the rAAV; coordinates in relation to bregma: *y*: −1.8 mm, *x*: 1.1 mm, *z*: −2.1 mm) or the ventral DG (*y*: - 2.9 mm, *x*: 2.5 mm, *z*: −2.3 to −2.9 mm) of homozygous P20-90 day-old SOM-Cre recombinase-expressing mice (Jackson Laboratories, Stock no. 013044). The rAAV expression cassette contained tdT and ChR2 between inverted incompatible two tandem loxP sites (rAAV-FLEX-ChR2-tdT). The detailed surgical procedure, injection and postoperative treatment of mice have been previously described (*Murray et al., 2011*; *Savanthrapadian et al., 2014*). We recently demonstrated the high selectivity of ChR2-tdT expression in SOMIs of SOM-Cre mice (*Savanthrapadian et al., 2014*). Acute transverse hippocampal or coronal septal slices (300 µm) were prepared from mice 14–18 days after injection. For excitation of ChR2-expressing fibers we followed two approaches: First, in slices of the MSvDB we applied blue light pulses (473 nm; 5 ms, 0.2 Hz, full field illumination; CoolLED system, UK). Second, for localized activation of ChR2-expressing SOM-fibers in DG slices we applied 1

Photon Laser-stimulation (0.5 ms, 0.2 Hz, 473 nm Laser; 0.5 µm diameter spot) to the outer molecular layer or close to the soma of the recorded cell. ChR2 expression can vary among mice. Thus, to directly compare the strength of light-mediated signals, we (a) step-wise increased the intensity of the applied light-pulse and obtained the maximal synaptic signal in recordings of the medial septum, and (b) in a subset of experiments in the DG and the medial septum we recorded light-induced IPSCs from two neurons / slice (*Figures 4* and *6*). Light-induced IPSCs were aligned to the onset of the light pulse and averaged from 20 to 40 traces. A signal was considered as IPSC if its negative peak amplitude exceeded three times the SD of the baseline noise.

For retrograde tracing of DG-SOMIs projecting to the medial septum, 0.5–1.0 µl RedRetroBeads (LumaFluor) were injected in the medial septum of anaesthetized mice (coordinates in relation to bregma; *y*: 1.01 mm, *x*: 1.1 mm, *z*: −4.1 mm; angle 15˚). The pipette was held in place for ~5 min and then retracted. Acute transversal hippocampal slices were prepared 3–8 days after injection for whole-cell recordings from labeled cell bodies in the DG.

## Data analysis

### Determination of passive membrane properties

Input resistance was measured under voltage-clamp conditions after application of a 25 ms, 10 mV voltage pulse. Membrane potentials reported in the text were not corrected for the junction potential. To determine $\tau_m$, 25 ms depolarizing subthreshold current pulses (10 mV) were applied in the voltage-clamp mode in the presence of 20 µM CNQX and 10 µM SR59931. Signals were averaged (10 traces) and the membrane capacitance ($C_m$) was determined by applying $\tau_m = C_m \times$ input resistance.

### Determination of active membrane properties

To define the properties of single action potentials, we injected depolarizing 1-s-long current pulses with step-wise increasing amplitude (step size: 50 pA). Only the first action potential during the initial 10 ms of current injection which crossed the spike threshold was considered in the analysis. The action potential threshold was defined as the first point in the voltage trajectory that exceeded a slope of 20 V/s (*Bekkers and Delaney, 2001*) during the rising phase of the action potential. Half-duration of individual action potentials was measured at the two points during the rise and decay phase halfway between threshold and peak. Maximum rise and decay time were defined as the maximal and minimal points of the first derivative of the voltage trajectory.

### Analysis of discharge patterns

Discharge frequencies were determined as the inverse of inter-spike intervals. Maximal discharge frequency was determined from current-frequency relationships (−100 to 500 pA current injections; step-size 50 pA; 1 s). Adaptation ratios are defined as the mean of the last three divided by the mean of the first three inter-spike intervals.

### Analysis of synaptic properties

Functional properties of uIPSCs were determined from averages of 30–50 traces including failures as previously described (*Savanthrapadian et al., 2014*). The synaptic latency was determined as the time interval between the steepest point in the rise of the presynaptic action potential and the onset of the postsynaptic uIPSC. The peak amplitude was defined as the maximum response within a 1–4 ms window following the presynaptic action potential. The decay of average uIPSCs was fitted with the sum of two exponentials [A exp(-t / $\tau_1$) + B exp(-t / $\tau_2$)], using a nonlinear least-squares fit algorithm; time constants are reported as amplitude-weighted means [$\tau_w = (A \tau_1 + B \tau_2) / (A + B)$]. Peak amplitudes were measured in relation to the preceding baseline in a 5–10 ms time window after the onset of stimulation. Data were analyzed using custom made software (Stimfit 0.13.2, https://code.google.com/p/stimfit/, courtesy of C. Schmidt-Hieber, University College London, UK).

*Statistical analysis* was performed using SigmaPlot 11 (Systat Software Inc., IL, USA) and custom-written scripts in MATLAB (MathWorks). All values are given as mean ± SEM. Statistical differences in the means of two samples were assessed by a two-tailed unpaired or a paired *t*-test for independent and related sample sets, respectively, if the samples were normally distributed as determined by the Shapiro-Wilk test. If the normality test failed, the non-parametric Mann-Whitney Rank Sum test or Wilcoxon Signed Rank test was employed. To compare three data sets for significant

differences (*Figure 6*), a Kruskal-Wallis one-way analysis of variance on Ranks was performed and in case that the normality test failed we used a Dunn's one-way analysis on variance. Significance levels are indicated as *p* values.

## Immunohistochemistry and morphology

Biocytin-filled cells slices were fixed in 4% paraformaldehyde overnight. After washing in phosphate buffer (PB, 0.1 M) and then phosphate-buffered solution (PBS, 0.025 M; ph = 7.3) containing 10% normal goat serum (NGS), slices were incubated with primary antibodies against SOM (monoclonal rabbit, 1:500, Peninsula Laboratories, San Francisco, USA) in PBS containing 5% goat-serum and 0.3% triton X-100 for 24 hr at room temperature. For visualization of SOM a secondary antibody goat anti-rabbit Cy3 (1:1000, Jackson ImmunoResearch, UK) was applied. The secondary antibody was administered together with streptavidin conjugated with Alexa Fluor 647 (1:500, Invitrogen, USA) in PBS and 0.1–0.3% triton X-100 for 24 hr at room temperature. Some preparations were subsequently incubated for 5 min in PBS containing 4´,6-diamidino-2-phenylindole (DAPI; 1:1000) to stain nuclei. Slices were finally washed in PBS then 0.1 M PB and embedded in Mowiol. Stained neurons were morphologically identified and examined for double-labelling using a LSM 710 confocal microscope (Zeiss; 10 x/20 x objective lenses; N.A. 0.5/0.8, respectively).

To quantify the identity of the target cells of DG-SOMIs in the medial septum of SOM-Cre mice, we performed in vivo immunohistochemistry 2–4 weeks after rAAV-FLEX-GFP (Penn Vector Labs, Philadelphia, USA) injection with methods described previously (*Sauer et al., 2015*). In brief, mice (P20-60) were anesthetized with 3% isoflurane (in 100% $O_2$) by inhalation and anaesthesia was continued with intraperitoneal injection of urethane (2 g/kg in 0.1 M PBS). Animals were perfused for 1–2 min with PBS, followed by 13 min PBS containing 4% PFA. Brains were carefully dissected, stored overnight in 4% PFA (4°C) and sliced (50–100 μm) with a DTK-1000 vibratome (Dosaka). After exposure of the slices to 4% NGS and 0.2% TritonX-100 (30 min), primary antibody incubations were performed overnight at 4°C in PBS containing 2% NGS and 0.1% TritonX-100. We used primary antibodies against SOM (1:500, rabbit, Peninsula Laboratories LLC.; 1:125, mouse, GeneTex, US), parvalbumin (PV, 1:1000, rabbit, SWANT, Switzerland) and choline acetyltransferase (ChAT, 1:1000, rabbit, Millipore) in combination with antibodies directed against gephyrin (1:1000, mouse, Synaptic Systems). Antibody binding was visualized with fluorophore-conjugated secondary antibodies (Cy3, 1:1000, rabbit, Jackson ImmunoResearch, UK; Alexa Fluor 647, 1:1000, mouse, Invitrogen, USA). Slices were mounted in Mowiol (Sigma-Aldrich, Germany) and images were obtained with a LSM710 confocal microscope (Zeiss) using 5x, 20x and 63x oil immersion objectives (N.A. 0.16, 0.8 and 1.4, respectively).

Pairs of biocytin filled cells were fixed in 2.5% paraformaldehyde, 1.25% glutaraldehyde and 15% picric acid in 0.1 M PB (12 hr, 4°C). After fixation, slices were treated with hydrogen peroxide (1%, 10 min) and rinsed in PB. After incubation in 10% and 20% sucrose solution, slices were snap-frozen in liquid nitrogen and thawed at room temperature. Then, they were transferred to PBS containing 1% avidin-biotinylated horseradish peroxidase complex (ABC; Vector Laboratories, USA) for ~12 hr. Slices were rinsed in PB and developed with 0.05% 3,3-diaminobenzidine tetrahydrochloride (DAB) and 0.01% hydrogen peroxide. Finally, they were rinsed several times in PB, embedded in Mowiol (Sigma-Aldrich, Germany) and identified using a Olympus Fluoview 1000 (63x oil immersion objective).

In this study, 39 SOMIs were intracellularly labelled and morphologically examined. From this group, 32 cells formed the basis of the here defined two contrasting SOMI types. The remaining 7 cells showed heterogeneous morphologies dissimilar to HIPP and HIL cells (*Figure 1—figure supplements 1* and *2*) and were excluded from this study. In a subset of 15 labeled interneurons (six HIPP and six HIL cells in *Figure 1—figure supplement 1*; one HIL cell in *Figure 3*; two other types of SOMIs in *Figure 1—figure supplement 2*), detailed reconstructions of the dendrites and axon were performed on the basis of image stacks using Fiji-ImageJ and the Simple Neurite Tracer plugin (http://fiji.sc/Simple_ Neurite_ Tracer, *Longair et al., 2011*). To define axonal distributions for each reconstructed cell, regions of interest (hilus, granule cell layer, inner and outer molecular layer) were defined manually. Reconstructed traces were transformed into a binary line-stack. Axonal and dendritic lengths for the manually defined region were quantified using L-Measure (http://cng.gmu.edu:8080/Lm/). Non-reconstructed SOMIs were visually identified by confocal-microscopy on the basis of axonal distributions. HIPP cell axons crossed the granule cell layer to arborize in the molecular layer forming a lateral stretch of axons predominantly in the outer molecular layer (*Hosp et al., 2014*). In contrast, HIL cells allocated

their axon to the hilus and seemed to avoid the granule cell and molecular layer. SOMIs projecting to the MSvDB and retrogradely labelled were identified based on their axonal distribution first in confocal image stacks and second after reconstruction of their axon using Fiji-ImageJ (13 cells; *Figure 2— figure supplement 1*). The axon was confined to the hilar region (length >2 mm). In two additional SOMIs, the axon was directed to the granule cell layer and visually detected in the inner molecular layer. None of the cells projected in the outer molecular layer.

## Cluster analysis

Hierarchical cluster analysis was performed for morphological and physiological properties of DG-SOMI cells using SPSS (V24.0; Chicago, Illinois) algorithms (Ward's method, Euclidian distance). Cluster analysis reveals dissimilarities between cells by calculating the intercellular distance in a multidimensional space, where each dimension corresponds to one of the quantified cellular parameters. Cluster tree diagrams group cells into classes with highest similarities. The larger the distance between classes the larger the difference among them. We used the Euclidean distance as dissimilarity measure and the Ward's minimum variance method as linkage procedure (*Dumitriu et al., 2007*; *Hosp et al., 2014*). Cluster analysis on morphological criteria has been performed on the basis of 12 reconstructed cells and on the basis of 6 morphological (total dendritic and axonal length, axonal length in the hilus, granule cell layer, inner and outer molecular layer) as well as six physiological properties (input resistance, $C_m$, $\tau_m$, max. discharge frequency, half-duration of single action potentials, decay of single action potentials depicted as triangles in *Figure 1*).

## Clarity

To directly demonstrate SOM-expressing axon trajectories originating in the DG and projecting to the medial septum, we injected rAAV-FLEX-GFP into the dorsal DG of 2 SOM-Cre mice as described above. Four weeks after injection mice were perfused with PBS followed with 120 ml 4% PFA. Brains were dissected and used for Clarity following published protocols (*Chung et al., 2013*; *Yang et al., 2014*). In brief, brains were washed in 0.1 M PBS at 4°C for 2–3 hr and incubated in hydrogel monomer solution for 24 hr. Air was replaced with nitrogen to induce the polymerization (37°C, 3–4 hr). Excess hydrogel was carefully removed *via* brief PBS washes. Polymerized brains were transferred into 50 ml conical tubes containing 4% SDS and 200 mM boric acid (in 0.1 M PBS, pH 8.5) and washed at 37°C during shaking for 3–4 weeks (two solution changes over the course of a day). The tissue was kept as whole mount and washed in PBS containing 0.1% Triton-X in 0.1 M PBS (37°C) for 24 hr. Finally, brains were embedded in refractive index matching (RIMs) imaging media (a custom economical recipe; *Yang et al., 2014*) and imaged using a Femto2D 2P-microscope (Zeiss 16x objective: N.A. 0.8). Stitching of individual images was performed using Fiji-ImageJ and XuvStitch (*Emmenlauer et al., 2009*). Images were 3D-reconstructed using Imaris imaging software (Bitplane).

## Acknowledgements

We thank Kerstin Semmler and Karin Winterhalter for technical assistance. We thank Dr. Imre Vida for comments on previous versions of the manuscript. We thank Thomas Hainmüller for training MY in whole-cell recordings, immunohistochemistry and data analysis and Dr. Ilka Diester for her support in establishing the Clarity technology. We thank Dr. Roland Nitschke (Life Imaging Center University of Freiburg) for his support in using the Imaris Software. This work was supported by the MOTIVATE program of the University of Freiburg (TM), grants from the VW-Foundation (Lichtenberg Professorship Award to MB), the Deutsche Forschungsgemeinschaft (FOR2143, MB), the Schram Foundation (MB) and the Brain-Links Brain-Tools, Cluster of Excellence of the Deutsche Forschungsgemeinschaft (EXC 1086, MB).

## Additional information

**Competing interests**

MB: Reviewing editor, *eLife*. The other authors declare that no competing interests exist.

## Funding

| Funder | Grant reference number | Author |
|---|---|---|
| Deutsche Forschungsgemeinschaft | FOR2143 | Marlene Bartos |
| Volkswagen Foundation | Lichtenberg Award | Marlene Bartos |

The funders had no role in study design, data collection and interpretation, or the decision to submit the work for publication.

## Author contributions

MY, Formal analysis, Methodology, Writing— editing, clarity, Optogentic experiments in the DG and medial septum, Retrograde labeling of cells, Reconstructions; TM, Data curation, Formal analysis, Validation, Synaptic plasticity experiments, Reconstructions, Retrograde labeling, Classification of SOMIs, DCG-IV experiments, Involved in the design of the study; CB, Validation, Plasticity experiments, Reconstructions, Data analysis, Revision; SS, Paired recordings and data analysis, Editing; LA-B, AF, PW, rAAV design, Production and validation; PA, Analysis and interpretation of plasticity and electrophysiological data, Revision; CE, Analysis and interpretation of optogentic experiments in the DG, Revision ; MB, Conceptualization, Supervision, Validation, Visualization, Writing—original draft, Project administration, Writing—review and editing

## Author ORCIDs

Marlene Bartos, http://orcid.org/0000-0001-9741-1946

## Ethics

Animal experimentation: All animal procedures were performed in accordance to national and european legislations (license no.: G-11/53; X-12/20D).

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
