## [Decision Letter]

Thank you for submitting your article "Somatostatin-positive interneurons in the dentate gyrus of mice provide local- and long-range septal synaptic inhibition" for consideration by *eLife*. Your article has been reviewed by two peer reviewers, and the evaluation has been overseen by Gary Westbrook as the Senior Editor and Reviewing Editor. The reviewers have opted to remain anonymous. The reviewers have discussed the reviews with one another and the Senior Editor has drafted this decision to help you prepare a revised submission.

Summary:

As you will see, both reviewers had a number of comments, some of which will require some additional data but both reviewers were interested in the work and thought you would be able to address these issues within approximately 2 months. Even if you cannot complete every requested experiment, we think this can be a strong paper if you focus your conclusions towards their strongest data sets. A major concern was that the classification parameters are not well described and many conclusions are based on observations in small datasets. The differences in the cell types, which is pitched as the key finding of the paper, seem somewhat overstated without addressing the issues raised. The full comments of the reviewers are below.

*Reviewer #1:*

The authors show that somatostatin-expressing interneurons (SOMs) in the dentate gyrus are comprised of two populations with distinct anatomical and functional properties. Little is known about SOMs in the DG, and they have generally been treated as a homogenous population, so a comprehensive analysis of their connectivity and function addresses an important issue. I find the main conclusion convincing but I have reservations about some of the specific analyses and conclusions.

1) Subsection “Layer-specific axon distributions define two contrasting DG-SOMI types”: Both cell types shown in Figure 1 appear to have substantially greater dendrite length than the average length reported in the text (~240 micron). Either this is an oversight, or the morphology of both cells in Figure 1 is not representative. It might be appropriate to include the other reconstructed cells as supplementary material if in fact there is variability in dendrite structure.

2) One of the most interesting observations is that HIP and HIL cells exhibit different long-term synaptic plasticity (Figure 3). However, the interpretation that MFs generate postsynaptic cell-type specific plasticity not fully convincing because the authors have not shown that all the synaptic input to HIPs and HILs are from MFs. Since mossy cells also innervate interneurons in the hilus (i.e. Larimer and Strowbridge, 2008), an alternative possibility is that there is presynaptic cell-type specificity that dictates the polarity of plasticity. EPSCs in HIP and HIL cells may have different sensitivity to DCG-IV (Figure 3—figure supplement 1), potentially suggesting a different degree of innervation by MFs and mossy cells. Such a difference in excitatory input would be important for understanding distinct network functions as well as the mechanism underlying plasticity.

Along the same lines, the authors suggest (in the Methods) that washout of intracellular components could explain a lack of plasticity in 5/11 HIPP cells and 5/13 HIL cells, but this explanation is not satisfying without additional evidence (is plasticity more likely to occur with high resistance pipettes or perforated patch?). An alternative explanation is that some fibers produce LTP and some LTD, such that the net effect depends on the combination of activated inputs.

3) The idea that SOMs provide both dendritic and perisomatic inhibition seems reasonable based on the axonal targeting of HIPPs and HILs shown in Figure 1. But it doesn't make sense to compare IPSPs across cell types (interneurons and GCs) with different passive membrane properties, since the faster time constant of interneurons will assure faster IPSP kinetics regardless of dendritic filtering (Figure 4). These experiments should be done in voltage clamp.

4) One important variable that was not specified in regards to Figure 6 is the timing of the experiments relative to the ChR2 injection, since expression level of ChR2 has a significant impact on the light-evoked recruitment of fibers. In the cluster-firing cell of panel D, the 2-ms light pulse appears to generate two presynaptic spikes (either in the same axon or different axons) since the IPSC has a reliable double peak but this is not seen in the other examples. This could suggest a higher level of ChR2 expression. To assure that presynaptic recruitment is the same across all cell types, the experiments must be performed during the same window of time after the viral injection (the Methods states only that experiments were performed > 2 weeks post injection.)

*Reviewer #2:*

This is an interesting study that highlights the functional connectivity of somatostatin positive GABAergic neurons within the dentate gyrus and long-range to the medial septum. The paper emphasizes the presence of two subpopulations of DG SOM inhibitory neurons: Hilar-perforant path associated (HIPP described previously) and hilus associated interneurons (HIL) which this paper categorizes for the first time. The HIPP neurons get inputs from granule cells and target the molecular layer while the HILs target the hilus and provide somatic inhibition to PV neurons and long range inputs to the medial septum.

The study uses modern methods like optogenetics, clarity and plasticity assays. The manuscript presents important data regarding target selective connectivity and functional properties of GABAergic neurons. Inhibition in the dentate gyrus is not as well characterized as other regions of the hippocampus, and the findings of the study are novel and relevant to the field. However, the study is not as thorough as previous work from the same group. The format of the paper is predominantly characterization based. The relatively small datasets, lack of details regarding classification criteria and evidence for the functional role of HIPP and HIL, limit the scope and impact of the study without additional supporting data.

Results in Figure 1, refer to 32 cells where 8 are considered HIPP and 24 as HIL; however axon length and distribution data is presented from only 4 reconstructed cells in each category. Was this because the rest of the cells did not fill completely? How were partially filled neurons categorized based on morphology (20 for HIL)? Please show reconstructions of more neurons and present clearly what criteria for classification were used. Based on results from Figure 1, with GIN and Som-Cre-tdTom mice there should be more data for reconstruction and quantification of axon length and distribution.

There are small differences in electrophysiological properties, and as stated in the paper the HIL and HIPP were mainly categorized based on morphology. It would be helpful to perform multivariate analyses (e.g. PCA, HCA) to show differences between different classes similar to Graves et al., 2012, Fuchs et al., 2016, McGarry et al., 2010., Ascoli et al., 2008 What is the resting membrane potential of these two types of neurons?

Results from the second paragraph/Figure 3, that "HILS form the major anatomical substrate for long-distance DG-spetal projections" is based on the result that the 75% of labelled cells present a "morphological characteristics similar to HILs". There is no detail about the morphological analysis of these cells. Furthermore, can the authors detail the electrophysiological properties (Rin, AP HD, max. AP Frequency) of these long-range projecting neurons to confirm that they look similar to what they found previously.

Were the recordings for Figure 3 performed in GIN or SOM Cre-Ai9 mice to specifically target the SOM interneurons? If not was the SOM identity verified user SOM counterstaining? The time course for measurement of LTD presented in Figure 3, is too short and unstable. Please provide data from longer recordings and greater N.

Please present paired pulse ratio in addition to the transmission failure rates. Also, provide a supplementary figure showing resting membrane potential, input resistance through the timecourse of the experiment. Previous studies on LTD typically exclude effects of run down using perforated patch recordings. Prior to LTD a basic characterization of basal synaptic transmission would be valuable. For example, as per figure A and B the starting baseline EPSC amplitude in the HIPP vs. HIL cells look considerably different (200 nA vs. 50 nA). Is this representative of the two groups? If not then perhaps the LTP expression is an outcome of the smaller starting amplitude for HIL. Could one plot a correlation between the starting EPSC amplitude and the degree of LTD or LTP in the groups? Again, additional data to support the classification (morphological/physiological) from the HIPP vs. HIL cells should be presented. These long-term recordings must have yielded very good cell fills for detailed reconstruction. Figure 3—figure supplement 1 does not specify if example traces are from HIPP or HIL.

The argument for GC dendrite targeting nature of the HIPP cells is weak. The size of the oml stimulation evoked response in both GC and IN groups is similar for the dendritic layer photostimulation. The small amplitude could reflect the attenuation down the dendritic tree but also differences in membrane time constants of the two populations (GC vs. INs). In Figure 4 it is unclear what the 3 bars in the plot represent. For comparing strength of synaptic connections in a slice with virally expressed ChR2, it is important to perform dual recordings from each cell type (1 GC and 1 IN from each slice at least). The same applies to Figure 6. See McGarry and Carter, 2016.

Data in Figure 7 is purely anatomical but it is important to use the same approach as Figure 6 to determine functional connectivity (ChR2 evoked synaptic responses in DG-SOMIs).

What is the cross talk between the 2 groups of SOM DGs themselves?

[Editors' note: further revisions were requested prior to acceptance, as described below.]

Thank you for resubmitting your work entitled "Somatostatin-positive interneurons in the dentate gyrus of mice provide local- and long-range septal synaptic inhibition" for further consideration at *eLife*. Your revised article has been favorably evaluated by Gary Westbrook (Senior Editor and Reviewing Editor) and two reviewers.

Although some of the issues raised in the original reviews have been addressed with further analysis and experiments, the reviewers still had concerns that will require your attention. We think the observations are interesting and for the most part the experiments are well done. However, in some cases the small sample size precludes some of the strongest conclusions put forth by the authors. Thus, the conclusions need to be toned down to match the experimental data. In that regard, the text of the manuscript will need editing throughout following careful attention to the below comments. The Senior Editor will assess your response in the revised version.

Major points:

1) Figure 1. One of the reviewers still favored testing a cluster analysis to better determine the subtypes of SOM-expressing interneurons. The correlation analysis suggested in Hosp et al., 2014 may not be informative but a PCA or cluster analysis would. Please consider this possibility or discuss in the paper whether or not it would be useful.

2) The physiology data shows marginally significant differences between groups but included non-reconstructed neurons. Please plot the electrophysiological properties of fully reconstructed neurons and mark them with special symbols. The reason for concern is based on the examples of cells displayed in Figure 3, where a confocal image was used to see the location of cutoff axon segment or direction of innervation to determine the class. These examples could pass for non-HIPP and non-HIL cells if their axons were only partially visible.

3) Figure 3. It would be helpful to show a longer pre-induction baseline and a longer post induction time course. The LTD effect (sampled at 15-20 mins for E and F) only appears at the 16-17 min. In addition, the DCG application timecourse (20-40 mins) should be included in Figure 3 to demonstrate whether the magnitude of plasticity was uncorrelated with DCG-IV sensitivity.

4) One of the reviewers had this additional suggestion regarding Figure 3 to which you should respond:

"There is a flaw in the experimental design of bath applying DCG post LTP or LTD induction and expression and comparing this to the effects of DCG IV in naïve slices from a different data set. The differential impact of DCG IV on HIPP and HIL cells is interesting. It would be interesting to test if the differential expression of plasticity is due to presynaptic property differences (influence of MF versus mossy cells/FFI). A better experiment to perform is to bath apply DCG IV prior to induction and have it constantly present during induction and expression – throughout the course of the plasticity experiment. This would reveal if the LTP/LTD is independent of target selective presynaptic plasticity differences. If the approach chosen by the authors is to be used, then one must compare the effects of DCG IV application pre and post plasticity induction with washout in between during induction."

5) Figure 4. The authors’ premise about synapse location is supported by the similar rise times of somatic and dendritic evoked IPSCs in granule cells, whereas somatic-evoked IPSCs in interneurons have a faster rise time than dendritic-evoked IPSCs. However, the amplitude data is less convincing as it will depend entirely on the number of activated axons and release probability. The conclusions would be strengthened by additional data and analysis to parse out the subtype-selective contributions. The paired recordings shown in Figure 4 are a good complement to results in 4C and 4D to further strengthen the point that IN receive inhibition from HIL; but it still does not exclude the possibility that GCs receive somatic inhibition. Similar paired recordings between HIL and GCs will be the only convincing evidence to back the conclusion. Thus, the authors must tone down the weakest conclusion. Specifically, this statement in Results – "Thus, HIPP cells provide dendritic inhibition onto GCs and interneurons whereas SOM+ axons in the hilus supply powerful perisomatic inhibition onto interneurons" – should be changed to "Thus, HIPP cells provide dendritic inhibition onto GCs and interneurons whereas SOM+ axons in the hilus supply powerful perisomatic inhibition onto interneurons."

6) Figure 7. The lack of functional analysis makes the conclusion from this anatomical analysis weak. We strongly suggest that this figure be deleted.

---

## [Author Response]

*Summary:*

*As you will see, both reviewers had a number of comments, some of which will require some additional data but both reviewers were interested in the work and thought you would be able to address these issues within approximately 2 months. Even if you cannot complete every requested experiment, we think this can be a strong paper if you focus your conclusions towards their strongest data sets. A major concern was that the classification parameters are not well described and many conclusions are based on observations in small datasets. The differences in the cell types, which is pitched as the key finding of the paper, seem somewhat overstated without addressing the issues raised. The full comments of the reviewers are below.*

We thank the Senior Editor for the opportunity to revise our study and thereby improve the scientific content of our manuscript. In response to both reviewers, we focused our revision on three major parts:

1) We increased the number of reconstructions of the two contrasting somatostatin-expressing interneuron (SOMI) types, the hilar-perforant path-associated interneurons (HIPP) and the hilar interneurons (HILs). This resulted in the new Figure 1—figure supplement 1 and Figure 1—figure supplement 2.We show now 6 reconstructed HIPP, 6 HIL cells and 2 SOMIs which do not fall in the HIPP and HIL cell classification. We improved the description of classification criteria of the two interneuron types in the Materials and methods (subsection “Immunohistochemistry and Morphology”, last paragraph) as well as the Results section (subsection “Layer-specific axon distributions define two contrasting DG-SOMI types”, last paragraph). We included additional qualitative and quantitative information on the passive and active membrane properties of the two SOMI types such as membrane resting potentials, membrane time constants, the slope of decay of a single action potential and phase plots from individual action potentials. This resulted in new graphs in Figure 1.

2) We increased number of experiments demonstrating long-lasting synaptic potentiation (LTP) in HIL cells and included a new set of plasticity experiments showing long-lasting depression (LTD) in HIPP cells with subsequent DCG-IV bath application which resulted in a revised Figure 3 and new Figure 3—figure supplement 1. We performed a coefficient-of variation (CV) analysis which together with the failure rate analysis indicates that both LTD and LTP of synaptic transmission onto HIPP and HILs, respectively, is expressed presynaptically. This resulted in a new Figure 3—figure supplement 3. We performed additional quantitative data analysis as requested by the reviewers and show that the input resistance of SOMIs does not alter after plasticity induction compared to baseline periods (subsection “Electrophysiology”, last paragraph). Additionally, we provide statistical evidence that LTD and LTP do not depend on the peak amplitude of EPSCs during baseline periods (subsection “Differential forms of synaptic plasticity at glutamatergic synapses targeting HIPP and HIL cells”). We added a new set of experiments showing that HIPP cells receive DCG-IV-sensitive excitatory inputs indicating their Mossy Fiber (MF)-mediated nature. Our data further show that HIL cells receive DCG-IV sensitive as well as insensitive excitatory inputs indicating that HIL cells receive in addition to MFs also inputs from other glutamatergic cells, very likely local Mossy cells (MCs). These results are shown in the new Figure 3—figure supplement 1. In response to reviewer 2 we performed additional 3 experiments showing that LTD is long-lasting, extending a recording period of 30 min. This shown in the new Figure 3—figure supplement 2.

3) We replaced all current-clamp experiments originally shown in Figure 4 by voltage-clamp experiments and demonstrate that optogenetic activation of SOMI-positive synaptic inputs targeting dendrites of granule cells (GCs) and dentate gyrus fast-spiking interneurons evoke small and slow IPSCs whereas activation of perisomatic SOMI-positive synapses evokes large and rapid IPSCs in fast-spiking interneurons but not in GCs (Figure 4). This further supports our initial conclusion that persiomatic SOMI-expressing synapses are largely formed at interneurons but not at GCs.

Finally, we performed additional qualitative and quantitative data analysis requested by the reviewers which we list in the point-by-point response to both reviewers below. After this rigorous revision we hope that our study can fulfil the high standards of *eLife*.

*Reviewer #1:*

*The authors show that somatostatin-expressing interneurons (SOMs) in the dentate gyrus are comprised of two populations with distinct anatomical and functional properties. Little is known about SOMs in the DG, and they have generally been treated as a homogenous population, so a comprehensive analysis of their connectivity and function addresses an important issue. I find the main conclusion convincing but I have reservations about some of the specific analyses and conclusions.*

*1) Subsection “Layer-specific axon distributions define two contrasting DG-SOMI types”: Both cell types shown in Figure 1 appear to have substantially greater dendrite length than the average length reported in the text (~240 micron). Either this is an oversight, or the morphology of both cells in Figure 1 is not representative. It might be appropriate to include the other reconstructed cells as supplementary material if in fact there is variability in dendrite structure.*

We thank the reviewer for spotting this typo. The total dendritic length for HIPP cells is 2500.4 ± 333.0 µm (6 cells) and for HIL cells 2547.3 ± 427.8 µm (6 cells). We increased the number of reconstructed cells from the original 4 / group to 6 / group shown in the new Figure 1—figure supplement 1. The last paragraph of the subsection “Layer-specific axon distributions define two contrasting DG-SOMI types” was altered accordingly.

*2) One of the most interesting observations is that HIP and HIL cells exhibit different long-term synaptic plasticity (Figure 3). However, the interpretation that MFs generate postsynaptic cell-type specific plasticity not fully convincing because the authors have not shown that all the synaptic input to HIPs and HILs are from MFs. Since mossy cells also innervate interneurons in the hilus (i.e. Larimer and Strowbridge, 2008), an alternative possibility is that there is presynaptic cell-type specificity that dictates the polarity of plasticity. EPSCs in HIP and HIL cells may have different sensitivity to DCG-IV (Figure 3—figure supplement 1), potentially suggesting a different degree of innervation by MFs and mossy cells. Such a difference in excitatory input would be important for understanding distinct network functions as well as the mechanism underlying plasticity.*

We thank the reviewer for pointing out the importance of input specificity. Indeed, the comments of the reviewer motivated us to perform additional experiments in which we positioned the extracellular stimulation pipette at the granule cell layer (gcl) to hilus border to excite both Mossy Fiber (MF) synapses originating from GCs and Mossy cells (MCs). After obtaining a control period we bath applied DCG-IV which has previously be shown to be a specific group II metabotropic glutamate receptors (mGluRs) agonist which specifically reduces synaptic transmission at MF terminals. We observed that evoked EPSCs in HIPP cells could be substantially blocked by DCGIV by 51.7 ± 3.9% (range 35 – 79%, 10 cells; Figure 3—figure supplement 3) indicating their MF-mediated nature. Excitatory signals in HIL cells were also diminished by DCG-IV, but to a lesser extent (by 32.6 ± 9.4%, 16 cells; Figure 3—figure supplement 3). In contrast to HIPP cells, the magnitude of the DCG-IV effect was highly variable in HIL cells (range 0 – 95%; Figure 3—figure supplement 3). These data indicate that HILs receive in addition to MFs, excitatory inputs of different origin, very likely MCs. Representative traces as well as the quantitative data analysis are shown in the new Figure 3—figure supplement 1. On the basis of these findings we performed additional plasticity experiments with subsequent bath-application of DCG-IV which are shown in the revised Figure 3. Indeed, all LTP and LTD experiments shown in Figure 3 contain DCG-IV bath-application at the end of the plasticity experiment (20-30 min after plasticity induction). Our data show that independent on the DCG-IV effect, thus independent on the nature of the presynaptic input, HIPP cells expressed predominantly LTD and HIL cells LTP (Spearman’s Rank-Order correlation between DCG-IV effect and magnitude of synaptic plasticity, P > 0.05; subsection “Differential forms of synaptic plasticity at glutamatergic synapses targeting HIPP and HIL cells”).

To determine the locus of LTD and LTP expression, we examined changes in the percentage of transmission failures and performed a coefficient-of-variation (CV) analysis following Malinow and Tsien 1990 (Nature 346:177). Our data indicate a presynaptic locus of plasticity expression. These data are shown in the new Figure 3—figure supplement 3and were added to the Results section, subsection “Synaptic plasticity at synapses targeting HIPP and HIL cells is presynaptically expressed”.

*Along the same lines, the authors suggest (in the Methods) that washout of intracellular components could explain a lack of plasticity in 5/11 HIPP cells and 5/13 HIL cells, but this explanation is not satisfying without additional evidence (is plasticity more likely to occur with high resistance pipettes or perforated patch?). An alternative explanation is that some fibers produce LTP and some LTD, such that the net effect depends on the combination of activated inputs.*

Due to the low yield of morphologically identified HIPP cells as well as the extremely challenging Gramicidin perforated-patch recordings, we decided to perform additional experiments with high resistance electrodes (Rs >10 MOhm) for >30 min in which LTD was induced in all 3 SOMIs (1 HIPP cell and 2 unidentified SOMIs; new Figure 3—figure supplement 2; see also reviewer 2 point 4). Moreover, our new data set on synaptic plasticity includes only experiments with DCG-IV bath- application at the end. In this new data set 2 out of 7 HIPP cells and 1 out of 12 HIL cells show no synaptic plasticity. Thus, our initial proposal that wash-out of intracellular components may cause a lack of plastic changes may not hold and was therefore removed from the Materials and methods section.

Our data show that the sign and magnitude of synaptic plasticity does not depend on the magnitude of the DCG-IV effect (subsection “Differential forms of synaptic plasticity at glutamatergic synapses targeting HIPP and HIL cells”). We conclude therefore, that, the sign of synaptic plasticity (LTD and LTP) is not dependent on the nature of the presynaptic glutamatergic inputs but the identity of the target cell.

*3) The idea that SOMs provide both dendritic and perisomatic inhibition seems reasonable based on the axonal targeting of HIPPs and HILs shown in Figure 1. But it doesn't make sense to compare IPSPs across cell types (interneurons and GCs) with different passive membrane properties, since the faster time constant of interneurons will assure faster IPSP kinetics regardless of dendritic filtering (Figure 4). These experiments should be done in voltage clamp.*

We followed the reviewer’s suggestion and repeated the entire set of experiments under voltage-clamp conditions (12 GCs, 9 interneurons). This new data set is shown in the renewed Figure 4 and supports our initial conclusions that SOMIs induce dendritic small and slow IPSCs at both interneuron and GC dendrites but perisomatic fast and strong inhibitory signals at target interneurons but not at target GCs. The Results section was revised accordingly.

*4) One important variable that was not specified in regards to Figure 6 is the timing of the experiments relative to the ChR2 injection, since expression level of ChR2 has a significant impact on the light-evoked recruitment of fibers. In the cluster-firing cell of panel D, the 2-ms light pulse appears to generate two presynaptic spikes (either in the same axon or different axons) since the IPSC has a reliable double peak but this is not seen in the other examples. This could suggest a higher level of ChR2 expression. To assure that presynaptic recruitment is the same across all cell types, the experiments must be performed during the same window of time after the viral injection (the Methods states only that experiments were performed > 2 weeks post injection.)*

We thank the reviewer for emphasizing the importance of the methodological information. All experiments have been performed 14-18 days after viral injection. To exclude differences in EPSC size based on varying expression profiles in presynaptic DG-SOMI fibers, we obtained data from at least two cell types / animal / septal slice in a subset of experiments (3 PVIs and 3 cholinergic cells / slice; 3 PVIs and 3 putative glutamatergic cells / slice). These data are shown representatively for 3 PVI and 3 putative glutamatergic cells in the new Figure 6—figure supplement 1. Thus, differences in the strength of light-evoked EPSCs cannot be explained by differences in the expression profile of ChR2 but rather by differences in the convergence of synaptic inputs. 5

The methodological information is added to Materials and methods subsection “Optophysiology”, and to the Results subsection “DG-SOMIs provide strong inhibition onto putative septal glutamatergic but weak inhibition onto 307 GABAergic and cholinergic cells”. Finally, we increased the number of experiments and added the new data points to the revised Figure 6.

*Reviewer #2:*

*This is an interesting study that highlights the functional connectivity of somatostatin positive GABAergic neurons within the dentate gyrus and long-range to the medial septum. The paper emphasizes the presence of two subpopulations of DG SOM inhibitory neurons: Hilar-perforant path associated (HIPP described previously) and hilus associated interneurons (HIL) which this paper categorizes for the first time. The HIPP neurons get inputs from granule cells and target the molecular layer while the HILs target the hilus and provide somatic inhibition to PV neurons and long range inputs to the medial septum.*

*The study uses modern methods like optogenetics, clarity and plasticity assays. The manuscript presents important data regarding target selective connectivity and functional properties of GABAergic neurons. Inhibition in the dentate gyrus is not as well characterized as other regions of the hippocampus, and the findings of the study are novel and relevant to the field. However, the study is not as thorough as previous work from the same group. The format of the paper is predominantly characterization based. The relatively small datasets, lack of details regarding classification criteria and evidence for the functional role of HIPP and HIL, limit the scope and impact of the study without additional supporting data.*

*Results in Figure 1, refer to 32 cells where 8 are considered HIPP and 24 as HIL; however axon length and distribution data is presented from only 4 reconstructed cells in each category. Was this because the rest of the cells did not fill completely? How were partially filled neurons categorized based on morphology (20 for HIL)? Please show reconstructions of more neurons and present clearly what criteria for classification were used. Based on results from Figure 1, with GIN and Som-Cre-tdTom mice there should be more data for reconstruction and quantification of axon length and distribution.*

We increased the number of reconstructed cells from 4 to 6 for each group and provide a new Figure 1—figure supplement 1 showing 6 reconstructions / group. The criterion for interneuron identification was the visual inspection of the axonal arborization pattern. HIPP cells showed axon fibers crossing the granule cell layer (gcl) and forming a lateral stretch of axonal projections predominantly in the outer half of the molecular layer. Some HIPP cells formed axon collaterals in the hilus as previously shown by our work (Savanthrapadian et al., 2014; J Neurosci 34:8197). In case of HIL cells the axon was predominantly located in the hilus and did not distribute in the molecular layer. Cells were identified on the basis of visual identification from confocal image stacks. Partly labelled cells, which did not allow an identification of HIPP or HIL, were excluded from the study. The classification criteria have been added to the Materials and methods subsection “Immunohistochemistry and Morphology”, last paragraph.

*There are small differences in electrophysiological properties, and as stated in the paper the HIL and HIPP were mainly categorized based on morphology. It would be helpful to perform multivariate analyses (e.g. PCA, HCA) to show differences between different classes similar to Graves et al., 2012, Fuchs et al., 2016, McGarry et al., 2010., Ascoli et al., 2008 What is the resting membrane potential of these two types of neurons?*

Following the suggestion of the reviewer we included information on the resting membrane potentials of the two SOMI types to the Results subsection “DG-SOMI types have different intrinsic membrane properties” and to Figure 1.

On the basis of the very distinct and significantly different morphological axonal distribution patterns of the two SOMI types and the small but significantly different physiological parameters (please see the added passive and active membrane properties in Figure 1), we believe that a cluster analysis as previously performed on dentate gyrus interneuron types (Hosp et al., 2014, Hippocampus, 24:189) will not improve our current results and therefore decided not to perform a cluster analysis. We hope that the reviewer can agree with our decision.

*Results from the second paragraph/Figure 3, that "HILS form the major anatomical substrate for long-distance DG-spetal projections" is based on the result that the 75% of labelled cells present a "morphological characteristics similar to HILs". There is no detail about the morphological analysis of these cells. Furthermore, can the authors detail the electrophysiological properties (Rin, AP HD, max. AP Frequency) of these long-range projecting neurons to confirm that they look similar to what they found previously.*

In Figure 2 we show one example of one projecting HIL cell. To improve the clarity on the classification criteria we define the identification parameters in the Results subsection “HIL but not HIPP cells form long-range connections to the medial septum”, last paragraph and in the Materials and methods subsection “Immunohistochemistry and Morphology”, last paragraph. We state that retrogradely labelled cells in the DG had axon fibers located in the hilus but not in the outer molecular layer and therefore were classified as HIL cells. This classification is based on reconstructions from retrogradely tagged and intracellularly labelled cells. Only neurons with an axonal length of >2 mm were considered in the analysis. This resulted in a total of 13 HILs and 2 cells with axon in the hilus and inner molecular layer. We show in the new Figure 2—figure supplement 1 two additionally reconstructed and projecting HIL cells.

The loading of cells with RedRetroBead markedly changed the membrane characteristics. The input resistance for example increased to 380.4 ± 44.4 MΩ markedly different to the one observed in control HIL cells with 186.7 ± 12.1 MΩ. Moreover, the maximal discharge frequency of retrogradely labelled cells was markedly lower with 56.0 ± 11 Hz than in controls with 141.5 ± 5.7 Hz. We therefore focused our analysis on axon localization only.

*Were the recordings for Figure 3 performed in GIN or SOM Cre-Ai9 mice to specifically target the SOM interneurons? If not was the SOM identity verified user SOM counterstaining? The time course for measurement of LTD presented in Figure 3, is too short and unstable. Please provide data from longer recordings and greater N.*

In all plasticity experiments shown in this study antibody labelling against SOM has been performed (GIN, n = 11 and SOM Cre-Ai9 mice, n = 8 cells). Examples for one HIPP and one HIL expressing LTD and LTP, respectively, and the respective antibody labelling against SOM has been newly included to Figure 3. Following the proposal of the reviewer on the stability of the recordings and the request of reviewer 1 to prove the nature of excitatory inputs onto DG-SOMIs by bath-applying DCG-IV at the end of all experiments (major criticism 2), we increased the total number of experiments from originally 3 to 7 HIPP cell and from the original 6 to 12 HIL cells with stable recordings for 20 min after plasticity induction and subsequent DCG-IV bath-application. In an additional data set of 1 HIPP and 2 non-identified DG-SOMIs which expressed LTD, we increased the recording time from 20 to 30 min with subsequent DCG-IV bath-application resulting in a total post-induction recording time of 40 min. The new data sets are shown in Figure 3 and the new Figure 3—figure supplement 2 and demonstrate that LTD is a stable, robust and long-lasting phenomenon.

*Please present paired pulse ratio in addition to the transmission failure rates.*

The number of paired pulses was unfortunately too low for a useful data analysis. However, we performed a coefficient-of-variation (CV) analysis on EPSC peak amplitudes during the time window of 15-20 min after plasticity induction normalized to baseline conditions and plotted the obtained data against the normalized EPSC amplitudes (mean EPSC amplitude after plasticity induction/mean EPSC amplitude at baseline conditions). The majority of data points obtained from LTP experiments were located at or above the identity line and from LTD experiments close to the identity line indicating a presynaptic locus of plasticity expression for both LTP and LTD. The CV analysis together with the significant decline in the percentage of failures in synaptic transmission in LTP but significant increase in failure rates in LTD experiments (15-20 min after induction) compared to baseline values support the presynaptic locus of plasticity expression. These data are shown in the new Figure 3—figure supplement 3 and have been added to the Results subsection “Synaptic plasticity at synapses targeting HIPP and HIL cells is presynaptically expressed”.

*Also, provide a supplementary figure showing resting membrane potential, input resistance through the timecourse of the experiment.*

All experiments have been performed in voltage-clamp conditions. Thus, continuous values on the resting membrane potential (Vrest) as a function of recording time cannot be provided. However, we measure Vrest regularly every 5 min and did not observe significant changes in its value. This is now mentioned newly in the last paragraph of the subsection “Electrophysiology”.

We followed the proposal of the reviewer and analyzed in a subset of the recorded cells the input resistance (Rin) over recording time (see Figure 8). We show (1) that Rin during the baseline period and 15-20 min after plasticity induction did not significantly change (LTD cells: baseline 326.3 ± 21.2 MOhm, 15-20 min after LTD expression: 323.0 ± 25.5 MOhm, 8 cells; HIL cells: 201.5 ± 19.2 MOhm, after LTP expression: 204.8 ± 14.6 MOhm, 9 cells; Figure 8).

Moreover, Rin was stable during the course of the recordings (Figure 8). We included this new information in the Materials and methods subsection “Electrophysiology”, last paragraph.

Author response image 1.The input resistance (Rin) of HIPP and HIL cells does not change over recording time and in dependence on the applied associative burst frequency (aBFS) stimulation for the induction of synaptic plasticity.(**A**) Rin was determined on the basis of 10 mV test pulses applied to the recorded cells throughout the experiment. The mean values are plotted for baseline periods (pre) as well as 15-20 min after the aBFS application (post). Rin was not significantly different between baseline periods and after plasticity induction (P > 0.05, t-Test). Circles connected by lines represent individual experiments (8 blue circles represent 5 HIPP cells showing synaptic plasticity included in Figure 3 of the manuscript plus 3 DG-SOMIs expressing LTD and shown in Figure 3—figure supplement 2 of the revised manuscript; 9 red circles represent 9 HIL cells expressing LTP and shown in Figure 3 of the manuscript). (**B**) Rin was plotted against the recordings time for a subset of HIPPs (4 cells) and HILs (8 cells). Note the stable Rin throughout the experiment.**DOI:**
http://dx.doi.org/10.7554/eLife.21105.018

*Previous studies on LTD typically exclude effects of run down using perforated patch recordings. Prior to LTD a basic characterization of basal synaptic transmission would be valuable. For example, as per figure A and B the starting baseline EPSC amplitude in the HIPP vs. HIL cells look considerably different (200 nA vs. 50 nA). Is this representative of the two groups? If not then perhaps the LTP expression is an outcome of the smaller starting amplitude for HIL. Could one plot a correlation between the starting EPSC amplitude and the degree of LTD or LTP in the groups? Again, additional data to support the classification (morphological/physiological) from the HIPP vs. HIL cells should be presented. These long-term recordings must have yielded very good cell fills for detailed reconstruction. Figure 3—figure supplement 1 does not specify if example traces are from HIPP or HIL.*

The mean EPSC peak amplitude in HIPP cells expressing LTD was 77.0 ± 19.6 pA during baseline periods. The mean EPSC peak amplitude in HIL cells expressing LTP was 74.6 ± 16.4 pA during baseline periods. The mean peak amplitudes were not significantly different between HIPP and HIL cells (P = 0.767, t-test). The magnitude of synaptic plasticity did not correlate with the amplitude of baseline EPSCs as determined with the Spearman’s Rank-Order correlation (P > 0.05 for both comparisons). This information is shown below (Figure 9) and was added to the Results subsection “Differential forms of synaptic plasticity at glutamatergic synapses targeting HIPP and HIL cells”. In accordance with the reviewer’s proposal we added representative confocal images of an intracellularly labelled HIPP and HIL cell and the respective SOM antibody labelling to Figure 3. Moreover, we show example traces during extracellular stimulation of glutamatergic inputs with subsequent bath-application of

DCG-IV from both HIL and HIPP cells in the revised Figure 3—figure supplement 1.

Author response image 2.The magnitude of synaptic plasticity does not depend on the peak amplitude of EPSCs during the baseline period.The percentage of synaptic plasticity was plotted against the peak amplitude of the mean EPSCs recorded during baseline periods prior to the application of the aBFS for plasticity induction. Circles represent individual experiments from HIPP (blue) and HIL (red) cells.**DOI:**
http://dx.doi.org/10.7554/eLife.21105.019

*The argument for GC dendrite targeting nature of the HIPP cells is weak. The size of the oml stimulation evoked response in both GC and IN groups is similar for the dendritic layer photostimulation. The small amplitude could reflect the attenuation down the dendritic tree but also differences in membrane time constants of the two populations (GC vs. INs).*

Following the request of the reviewer, we repeated the experiments under voltage-clamp conditions and observed similar qualitative results. We recorded IPSCs with small amplitudes upon blue light pulse application close to the dendrites of both fast-spiking INs and GCs and IPSCs with large amplitudes and fast time courses in INs but not in GCs. These data therefore provide evidence of perisomatic SOMI-mediated inhibitory synapses in INs but not GCs. The new set of data have been added to the revised Figure 4. Please view also response to reviewer 1, point 3.

*In Figure 4 it is unclear what the 3 bars in the plot represent. For comparing strength of synaptic connections in a slice with virally expressed ChR2, it is important to perform dual recordings from each cell type (1 GC and 1 IN from each slice at least). The same applies to Figure 6. See McGarry and Carter, 2016.*

Figure 4 show representative data from 2 HIL-HIL and 2 HIL-BC paired recordings. The three bars represent the synaptic latency (lat), the 20-80% rise time (rise) and the decay time constant (τ) of unitary IPSCs. We improved the figure legend to enhance the clarity on the presented data.

Recordings have been performed from 12 GCs and 6 INs, from which 3 GCs and 3 INs were recorded in the same slice. The qualitative and quantitative data in simultaneous IN and GC recordings were similar to those ones recorded from individual INs and GCs/slice. We agree that the intensity of ChR2 expression can vary among injected mice. However, recordings have been always performed from GCs and INs of the same injected mouse. Moreover, we compared dendritic vs. perisomatic light-plus mediated IPSCs of the same recorded cell. We therefore strongly believe that differences in the peak amplitude and time course of IPSCs evoked by light-mediated ChR2- activation close to the soma vs. dendrite in INs as well as GCs cannot be explained inter-slice variabilities but differences in the location of input synapses. Finally, to exclude variabilities in IPSC size due differences in viral expression intensity, data shown on Figure 6 have been obtained in a subset of recordings from 2 cell types / slice (3 PVI plus 3 putative glutamatergic cell; 3 PVI plus 3 ChAT cells). This is now explicitly stated in the Materials and methods subsection “Optophysiology”, second paragraph, in the Results subsection “DG-SOMIs provide local dendritic and perisomatic inhibition onto target cells” and “DG-SOMIs provide strong inhibition onto putative septal glutamatergic but weak inhibition onto GABAergic and cholinergic cells”, last paragraph.

*Data in Figure 7 is purely anatomical but it is important to use the same approach as Figure 6 to determine functional connectivity (ChR2 evoked synaptic responses in DG-SOMIs).*

We thank the reviewer for emphasizing this important experiment which we indeed tried. We injected rAAV-ChR2-tdT in the medial septum of PV-Cre mice. Four weeks after viral expression we performed whole-cell recordings from ~100 hilar cells in acute hippocampal slice preparations to evoke light pulse-mediated IPSCs in target cells of the DG. However, we could not record any IPSCs. We conclude that the expression intensity of ChR2 over long distances is not as strong as in the opposite direction from DG-SOMIs projecting to the medial septum. Although we would have loved to have these data, we decided not to continue this set of experiments and hope that the reviewer can support our decision.

*What is the cross talk between the 2 groups of SOM DGs themselves?*

This is a very interesting question. In Figure 4 we show that HIL cells are mutually interconnected (2 HIL-HIL pairs) but also target basket cells (2 HIL-BC pairs). Detailed information on the connectivity and synaptic properties among SOMI types, however, are very difficult to obtain and therefore, the focus of an independent study, which we currently follow.

[Editors' note: further revisions were requested prior to acceptance, as described below.]

*Although some of the issues raised in the original reviews have been addressed with further analysis and experiments, the reviewers still had concerns that will require your attention. We think the observations are interesting and for the most part the experiments are well done. However, in some cases the small sample size precludes some of the strongest conclusions put forth by the authors. Thus, the conclusions need to be toned down to match the experimental data. In that regard, the text of the manuscript will need editing throughout following careful attention to the below comments. The Senior Editor will assess your response in the revised version.*

*Major points:*

*1) Figure 1. One of the reviewers still favored testing a cluster analysis to better determine the subtypes of SOM-expressing interneurons. The correlation analysis suggested in Hosp et al., 2014 may not be informative but a PCA or cluster analysis would. Please consider this possibility or discuss in the paper whether or not it would be useful.*

We performed the requested cluster analysis, which is newly included in Figure 1. The corresponding text was inserted in the Results subsection “DG-SOMI types have different intrinsic membrane properties”, last paragraph and a new chapter in the Materials and methods subsection”Cluster analysis”. The cluster analysis was performed by including morphological (axonal and dendritic length, percent distribution of the axon in the hilus, granule cell layer, inner and outer molecular layer; 6 parameters), and physiological properties (input resistance, membrane resting potential, membrane time constant, action potential half-duration, decay time course of single action potentials, maximal discharge frequency; 6 parameters summarized in Materials and methods) of the reconstructed SOMIs depicted as triangles in the Figure 1. This analysis confirmed our prediction that the recorded DG-SOMIs fall into two distinct groups, HIPP and HIL cells, on the basis of their axon location and intrinsic membrane properties. Due to space limits we moved the representative single action potential and the corresponding phase plot from a HIPP and a HILL cell to the new Figure 1—figure supplement 3.

*2) The physiology data shows marginally significant differences between groups but included non-reconstructed neurons. Please plot the electrophysiological properties of fully reconstructed neurons and mark them with special symbols. The reason for concern is based on the examples of cells displayed in Figure 3, where a confocal image was used to see the location of cutoff axon segment or direction of innervation to determine the class. These examples could pass for non-HIPP and non-HIL cells if their axons were only partially visible.*

We included the requested symbols in Figure 1. We apologize for the low quality of the figure leaving the reviewers with the impression that the axon was cut. We show now instead of the confocal images the morphological reconstructions of the two SOMI types confirming that they are a representative HIPP and a HIL cell.

*3) Figure 3. It would be helpful to show a longer pre-induction baseline and a longer post induction time course. The LTD effect (sampled at 15-20 mins for E and F) only appears at the 16-17 min. In addition, the DCG application timecourse (20-40 mins) should be included in Figure 3 to demonstrate whether the magnitude of plasticity was uncorrelated with DCG-IV sensitivity.*

We included the time course of the pre-control of 10 min in the revised Figure 3. We show the time course of the DCG-IV effect for individual experiments (LTD and LTP) in Figure 3. Due to the high heterogeneity of the DCG-IV effect among HIL cells we included a graph showing the DCG-IV effects for every individual plasticity experiment in Figure 3 (left). Moreover, we included a new plot in Figure 3 (right) showing that the degree of LTD/LTP and the magnitude of the DCG-IV effect are not correlated. We would like to emphasize that a significant reduction in the mean synaptic transmission (LTD) was expressed in the time window of 10-15 min as well as 15-20 min post induction compared to pre-induction periods shown in Figure 3 (right).

We show post-induction time courses for up to 21 min in Figure 3. We cannot provide longer time courses for this set of experiments. However, we show in Figure 3—figure supplement 2 the time course of LTD (3 SOMIs) for up to 30 min after plasticity induction with subsequent DCG-IV bath- application (total recording time 45 min). These experiments demonstrate that LTD in SOM-positive interneurons is stable and remains for long periods of time (>20 min after plasticity induction).

*4) One of the reviewers had this additional suggestion regarding Figure 3 to which you should respond:*

*"There is a flaw in the experimental design of bath applying DCG post LTP or LTD induction and expression and comparing this to the effects of DCG IV in naïve slices from a different data set. The differential impact of DCG IV on HIPP and HIL cells is interesting. It would be interesting to test if the differential expression of plasticity is due to presynaptic property differences (influence of MF versus mossy cells/FFI). A better experiment to perform is to bath apply DCG IV prior to induction and have it constantly present during induction and expression – throughout the course of the plasticity experiment. This would reveal if the LTP/LTD is independent of target selective presynaptic plasticity differences.*

We included in Figure 3 (left) the magnitude of the mean DFG-IV effect obtained from all individual synaptic plasticity experiments. The mean DCG-IV effect in HIPP and HIL cells (Figure 3, left) after the induction of plasticity was not significantly different from the ones observed in naive slices (P > 0.05, t- test; Figure 3—figure supplement 1; circles represent experiments in naive slices and triangles represent plasticity experiments). This information is now stated in the corresponding figure legend. Moreover, we include a plot in Figure 3 (right) demonstrating that the magnitude of synaptic plasticity (LTD and LTP) was uncorrelated with the extent of the DCG-IV effect on synaptic transmission. These data as a whole suggest that the magnitude of the observed plasticity changes (LTD/LTP) were independent on the nature of the input synapse.

Excitatory inputs that expressed LTD in HIPP cells were consistently DCG-IV sensitive (~52% block) pointing to their mossy fiber-mediated nature. In contrast, excitatory inputs in HIL cells showed a milder mean DCG-IV sensitivity (~32% block) with a high heterogeneity in the individual DCG-IV effects (range -2.4 to 96% blocking effect) suggesting that in addition to mossy fibers, different glutamatergic inputs, very likely originating from mossy cells, may contribute to synaptic plasticity. To test whether these additional glutamatergic inputs may express synaptic plasticity, we followed the proposal of the reviewer and performed an additional set of experiments in which we bath-applied DCG-IV before establishing the whole-cell recording of an SOMI and continued to apply DCG-IV throughout the entire experiment. Our data show that synaptic potentiation could still be induced (LTP by 144.9 ± 5.2%; 4 SOMIs; Figure 10). Indeed, LTP reached values similar to experiments in which EPSCs were markedly blocked by DCG-IV >20 min after LTP expression indicating their mossy fiber-mediated nature (Figure 3 of the main manuscript). These data as a whole support our hypothesis that synaptic plasticity in SOMIs is independent of the nature of the synaptic input and can be induced at mossy fiber and other types of glutamatergic synapses, very likely mossy cell inputs.

Author response image 3.Long-lasting potentiation can be induced at glutamatergic inputs targeting somatostatin-expressing interneurons (SOMIs) in the presence of the group II mGluR agonist DCG-IV.EPSCs were evoked by extracellular stimulation with a pipette positioned at thegranule cell layer to hilus border. Plot summarizes the time course of EPSC peak amplitudes evoked at glutamatergic input synapses targeting SOMIs before and after associative pairing as indicated by the arrow. EPSCs were averaged over 30 sec intervals and normalized to baseline values. Long-term potentiation (LTP) was determined 15-20 min after plasticity induction. Note, DCG-IV was bath-applied throughout the entire experiment to block mossy fiber-mediated EPSCs (4 SOMIs). These data together with our finding that LTP can be induced at glutamatergic inputs onto SOMIs that are strongly DCG-IV sensitive and therefore mediated by mossy fibers (blocking effect >40%; see Figure 3 left in the main manuscript), suggests that LTP induction in SOMIs is independent on the nature of the input synapse.**DOI:**
http://dx.doi.org/10.7554/eLife.21105.020

*If the approach chosen by the authors is to be used, then one must compare the effects of DCG IV application pre and post plasticity induction with washout in between during induction."*

The proposed experiments require the recording of a baseline period, followed by DCG-IV bath- application with plasticity induction, the subsequent wash-out of the drug and the recording of synaptic plasticity. A full recovery of synaptic transmission at mossy fiber synapses requires a long wash-out time (approximately 25 min). We are concerned that after DCG-IV washout residual effects of the drug on intracellular signaling cascades might generate unexpected or variable effects on synaptic release or on the expression of synaptic plasticity. Therefore, we fear that the results obtained from the proposed experiments might not provide a reliable answer to the reviewer’s question. The most convincing experiment would be the repetition of the here presented plasticity experiments in a pair configuration (GC-SOMI vs. mossy cell-SOMI pairs). However, these experiments are highly challenging due to the difficulty obtaining these pairs. We would therefore propose to tone down our conclusion at the end of the first paragraph of the subsection “Differential forms of synaptic plasticity at glutamatergic synapses targeting HIPP and HIL cells”, with the sentence, ‘…suggesting that LTP was induced at mossy fiber terminals and at other glutamatergic synapses, very likely those originating from mossy cells…’.

*5) Figure 4. The authors’ premise about synapse location is supported by the similar rise times of somatic and dendritic evoked IPSCs in granule cells, whereas somatic-evoked IPSCs in interneurons have a faster rise time than dendritic-evoked IPSCs. However, the amplitude data is less convincing as it will depend entirely on the number of activated axons and release probability. The conclusions would be strengthened by additional data and analysis to parse out the subtype-selective contributions. The paired recordings shown in Figure 4 are a good complement to results in 4C and 4D to further strengthen the point that IN receive inhibition from HIL; but it still does not exclude the possibility that GCs receive somatic inhibition. Similar paired recordings between HIL and GCs will be the only convincing evidence to back the conclusion. Thus, the authors must tone down the weakest conclusion. Specifically, this statement in Results – "Thus, HIPP cells provide dendritic inhibition onto GCs and interneurons whereas SOM+ axons in the hilus supply powerful perisomatic inhibition onto interneurons" – should be changed to "Thus, HIPP cells provide dendritic inhibition onto GCs and interneurons whereas SOM+ axons in the hilus supply powerful perisomatic inhibition onto interneurons."*

We fully agree and follow the proposal of the reviewer by toning down our statement at the end of the first paragraph of the subsection “DG-SOMIs provide local dendritic and perisomatic inhibition onto target cells”. (‘Thus, HIPP cells provide dendritic inhibition onto GCs and interneurons whereas SOM^+^ axons in the hilus seem to supply powerful perisomatic inhibition onto interneurons’).

*6) Figure 7. The lack of functional analysis makes the conclusion from this anatomical analysis weak. We strongly suggest that this figure be deleted.*

We deleted the anatomical analysis of this figure and the corresponding Results and Methods section. We kept, however, the schematic illustration summarizing the proposed synaptic connections among the DG cells including DG-SOMIs.